# Transferred mitochondria accumulate reactive oxygen species, promoting proliferation

**Chelsea U Kidwell[1†], Joseph R Casalini[1†], Soorya Pradeep[2], Sandra D Scherer[3], Daniel Greiner[1], Defne Bayik[4], Dionysios C Watson[4,5,6], Gregory S Olson[7], Justin D Lathia[4], Jarrod S Johnson[8], Jared Rutter[1,9,10], Alana L Welm[3], Thomas A Zangle[2,10], Minna Roh-Johnson[1,10]***

[1]Department of Biochemistry, University of Utah School of Medicine, Salt Lake City, United States; [2]Department of Chemical Engineering, University of Utah, Salt Lake City, United States; [3]Department of Oncological Sciences, Huntsman Cancer Institute, University of Utah, Salt Lake City, United States; [4]Department of Cardiovascular and Metabolic Sciences, Lerner Research Institute, Cleveland Clinic, Case Western Reserve University, Cleveland, United States; [5]University Hospitals Cleveland Medical Center, Cleveland, United States; [6]School of Medicine, Case Western Reserve University, Cleveland, United States; [7]Medical Scientist Training Program, University of Washington, Seattle, United States; [8]Division of Microbiology & Immunology, Department of Pathology, University of Utah School of Medicine, Salt Lake City, United States; [9]Howard Hughes Medical Institute, University of Utah School of Medicine, Salt Lake City, United States; [10]Huntsman Cancer Institute, University of Utah, Salt Lake City, United States

**\*For correspondence:**
roh-johnson@biochem.utah.edu

[†]These authors contributed equally to this work

**Competing interest:** The authors declare that no competing interests exist.

**Abstract:** Recent studies reveal that lateral mitochondrial transfer, the movement of mitochondria from one cell to another, can affect cellular and tissue homeostasis. Most of what we know about mitochondrial transfer stems from bulk cell studies and have led to the paradigm that functional transferred mitochondria restore bioenergetics and revitalize cellular functions to recipient cells with damaged or non-functional mitochondrial networks. However, we show that mitochondrial transfer also occurs between cells with functioning endogenous mitochondrial networks, but the mechanisms underlying how transferred mitochondria can promote such sustained behavioral reprogramming remain unclear. We report that unexpectedly, transferred macrophage mitochondria are dysfunctional and accumulate reactive oxygen species in recipient cancer cells. We further discovered that reactive oxygen species accumulation activates ERK signaling, promoting cancer cell proliferation. Pro-tumorigenic macrophages exhibit fragmented mitochondrial networks, leading to higher rates of mitochondrial transfer to cancer cells. Finally, we observe that macrophage mitochondrial transfer promotes tumor cell proliferation in vivo. Collectively these results indicate that transferred macrophage mitochondria activate downstream signaling pathways in a ROS-dependent manner in cancer cells, and provide a model of how sustained behavioral reprogramming can be mediated by a relatively small amount of transferred mitochondria in vitro and in vivo.

## Editor's evaluation

This important work demonstrates that the transfer of dysfunctional mitochondria stimulates proliferation in recipient cancer cells by serving as a signal to induce reactive oxygen species production that in turn activates signaling pathways that control cell cycle. Compelling cell biology assays

including rigorous microscopy with elegant reporters track the function and fate of transferred mitochondria in recipient cells. The work is relevant to the study of mitochondria, cancer, and immune cells and will be of broad interest to cell biologists and biochemists.

## Introduction

It has been previously described that mitochondria can undergo lateral transfer between cells (*Torralba et al., 2016*; *Antanavičiūtė et al., 2014*; *Lou et al., 2012*; *Rebbeck et al., 2011*; *Tan et al., 2015*; *Wang and Gerdes, 2012*; *Wang and Gerdes, 2015*; *Lampinen et al., 2022*). Mitochondria are dynamic organelles, known to provide energy for the cell, but more recently shown to have a variety of additional essential cellular functions (*Zong et al., 2016*). In animal models, a series of seminal studies revealed that cancer cells void of mitochondrial DNA still form tumors by obtaining mitochondria from stromal cells, thereby restoring cancer cell mitochondrial function, cellular respiration, and tumor formation (*Tan et al., 2015*; *Dong et al., 2017*). Other experiments suggest that mitochondrial transfer not only restores bioenergetics, but can alter the metabolic state of recipient cells (*Brestoff et al., 2021*; *Nicolás-Ávila et al., 2020*; *Phinney et al., 2015*; *Saha et al., 2022*; *Crewe et al., 2021*; *Korpershoek et al., 2022*; *Liu et al., 2022*; *van der Vlist et al., 2022*; *Yang et al., 2022*; *Liu et al., 2021*), allowing recipient cells to adapt to stressors or changes in the environment, prompting the development of methods targeting mitochondrial dysfunction in disease (*Patel et al., 2023*; *Caicedo et al., 2015*). Although these studies elegantly demonstrate that mitochondrial transfer alters recipient cellular behavior, many aspects of this process remain unclear. For instance, the rescue of cellular function is commonly attributed to enhanced mitochondrial energetic or metabolic profiles; however, the fate and function of transferred mitochondria in recipient cells are under-explored. Furthermore, it is unclear how cells respond to laterally transferred mitochondria if the recipient cells already have a fully functioning mitochondrial network, and in particular, if the transferred mitochondria only comprise a small subset of the overall mitochondrial network in the recipient cell.

Given that metastasis is a low-frequency event and is the consequence of changes in cellular behavior on the single-cell level, we aimed to examine the function and behavior of transferred mitochondria within individual recipient cells that have functioning endogenous mitochondrial networks. Using a combination of in vitro high-resolution microscopy, optogenetics, imaging flow cytometry, and in vivo tumor models, we demonstrate a previously undescribed mechanism of mitochondrial transfer-associated cellular reprogramming. Collectively, our data explain how a relatively small amount of transferred mitochondria can impact cellular behavior in the recipient cell with fully functioning endogenous mitochondria – Transferred macrophage mitochondria in cancer cells are *dysfunctional*, ROS accumulates at the site of transferred mitochondria, promoting ERK-mediated cancer cell proliferation.

## Results

### Cancer cells with macrophage mitochondria exhibit increased proliferation

We previously reported that macrophages transfer cytoplasmic contents to cancer cells in vitro and in vivo (*Roh-Johnson et al., 2017*), and hypothesized that a macrophage/cancer cell system would be ideal for probing mitochondrial transfer in cells with functioning mitochondrial networks. Our studies employed blood-derived human macrophages and a human breast cancer cell line, MDA-MB-231 (231 cells), stably expressing a mitochondrially localized mEmerald or red fluorescent protein (mito-mEm or mito-RFP, respectively; *Figure 1a*). We observed mitochondrial transfer from macrophages to 231 cells using live cell confocal microscopy (*Figure 1b*, arrowheads) and flow cytometry (*Figure 1c–d*; flow cytometry scheme in *Figure 1—figure supplement 1a*). Control gates were set to 0.2%, based on confirmation of mitochondrial transfer by FACS-isolation of distinct mEmerald+ populations (see methods for more information). With these methods, a range of transfer efficiencies were observed, which we attribute to donor-to-donor variability (*Figure 1d*), yet mitochondrial transfer was consistently observed in 231 cells, as well as to another breast cancer line, MDA-MB-468, and a melanoma cell line, A375 (*Figure 1—figure supplement 1b*). To determine whether macrophage

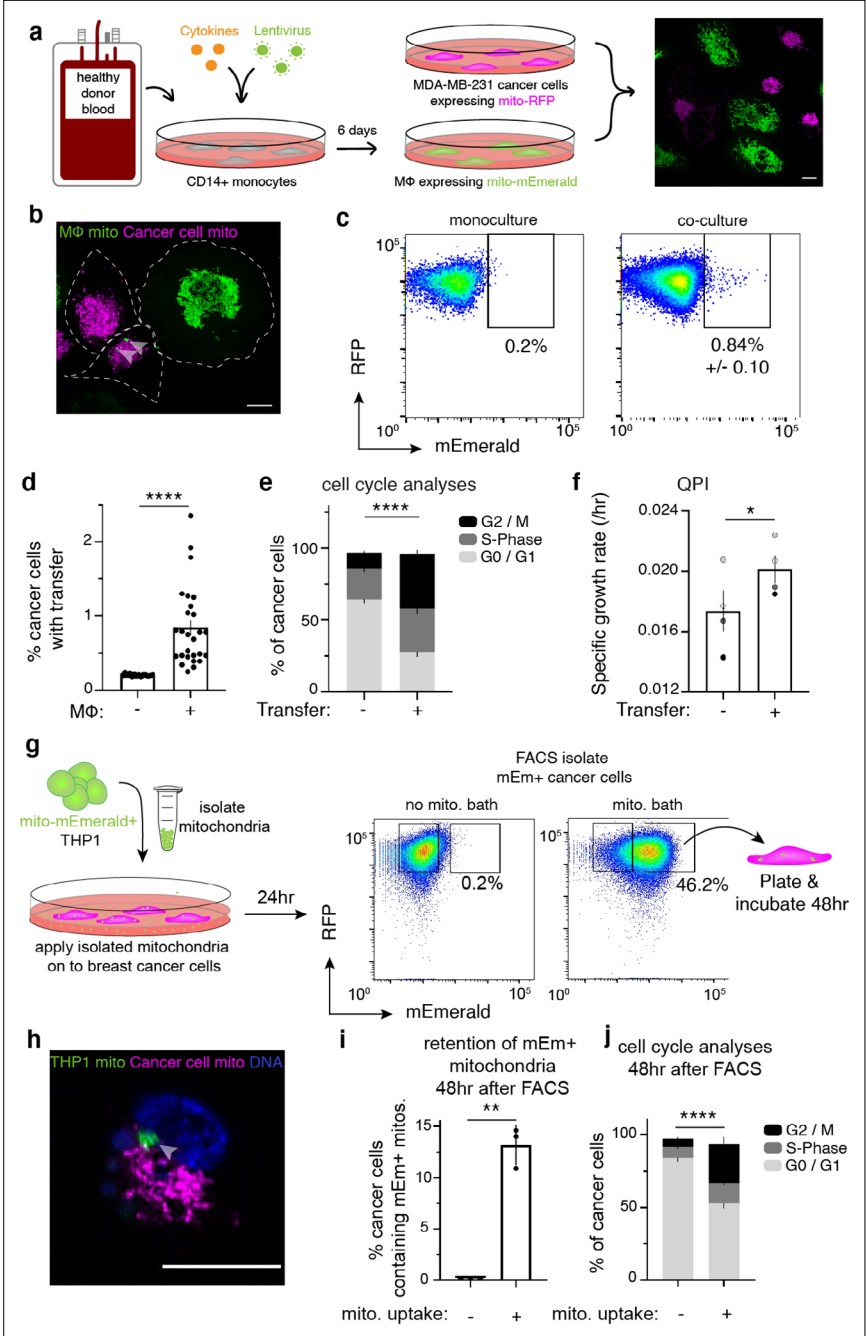

**Figure 1.** Cell-contact-mediated transfer of macrophage mitochondria leads to increased cancer cell proliferation. (**a**) CD14+ monocytes harvested from human blood are transduced and differentiated for 6 days. Mito-mEm +macrophages (green) are co-cultured with MDA-MB-231 cells (231 cells) expressing mito-RFP (magenta; right image). (**b**) Confocal image showing transferred mitochondria (green, arrowhead) in a 231 cell (magenta, cell outline in white). (**c**) Representative flow cytometry plots depicting mitochondrial transfer (black box) within a population of co-cultured mito-RFP+ 231 cells (right) compared to monoculture control (left) with background level of mEmerald (mEm) fluorescence set at 0.2%. (**d**) Aggregate data of mitochondrial transfer rates across macrophage donors. Each data point represents one replicate (N=14 donors). (**e**) Analysis of proliferative capacity by quantifying Ki-67 levels and DNA content in co-cultured 231 cells after 24 hr. Percentage of cancer cells within a specific cell cycle phase with or without transfer is shown. A significantly different percent of recipient cells occupies G2/M (black) phases of the cell cycle compared to non-recipient cells (N=4 donors; statistics for G2/M only). (**f**) Co-cultured recipient 231 cells have a significantly higher specific growth rate compared to non-recipients (N=60 cells (control), 115 (recipient) over 4 donors indicated as shades of gray). (**g**) Schematic of mitochondrial

*Figure 1 continued on next page*

*Figure 1 continued*

isolation and bath application on MDA-MB-231 cells. Mitochondria are isolated from mito-mEmerald expressing THP-1 monocytes and bath applied at 20–30 μg/mL for 24 hr. Cancer cells which had taken up mEm+ mitochondria are then FACS-isolated and plated for 48 hr for further analyses. (**h**) Representative confocal image showing mito-RFP-expressing 231 cell (magenta) that had taken up macrophage mitochondria (green, grey arrow). (**i**) 48 hr after FACS-isolating 231 cells with macrophage mitochondria, flow cytometry was used to determine percent of daughter cells which still contain mEm+ mitochondria. N=3 biological replicates. (**j**), Cell cycle analysis of daughter cells 48 hr after FACS-isolation of 231 cells that had taken up macrophage mitochondria. N=3 biological replicates. For all panels, standard error of the mean (SEM) is displayed and scale bars are 10 μm. Mann-Whitney (**d**), two-way ANOVA (**e, j**), Welch's t-test (**f, i**), *p<0.05; **p<0.01; ****p<0.0001.

The online version of this article includes the following figure supplement(s) for figure 1:

**Figure supplement 1.** Macrophages transfer mitochondria to cancer cells.

**Figure supplement 2.** Cancer cells will macrophage mitochondria exhibit increased proliferation.

**Figure supplement 3.** Mitochondrial transfer leads to sustained increased growth rate in daughter cancer cells.

mitochondrial transfer was unique to cancer cells, we tested a non-malignant breast epithelial cell line, MCF10A. We observed reduced mitochondrial transfer efficiencies to MCF10A cells, with no significant differences compared to control (*Figure 1—figure supplement 1c*), suggesting that macrophages exhibit higher mitochondrial transfer efficiencies to malignant cells. Transferred mitochondria contain a key outer mitochondrial membrane protein, TOMM20 (*Figure 1—figure supplement 1d*, arrowhead) and mitochondrial DNA (*Figure 1—figure supplement 1e*, arrowhead), suggesting that intact organelles are transferred to 231 cells. To better define the requirements for transfer, we performed trans-well experiments in which we cultured 231 cells either physically separated from macrophages by a 0.4 μm trans-well insert or in contact with macrophages (scheme in *Figure 1—figure supplement 1f*), or with conditioned media (*Figure 1—figure supplement 1g, h*). These data showed that mitochondrial transfer increased dramatically under conditions where 231 cells could contact macrophages directly (*Figure 1—figure supplement 1g and h*). Taken together, these results suggest that macrophage mitochondrial transfer to cancer cells likely requires cell-to-cell contact. Furthermore, while mitochondrial transfer may not be unique to cancer cells, macrophages transfer mitochondria to cancer cells at higher frequencies. Thus, due to the low rates of mitochondrial transfer across macrophage donors (0.84%, *Figure 1d*), we subsequently took advantage of single-cell, high-resolution approaches – rather than bulk approaches – to follow the fate and functional status of transferred mitochondria.

To determine the effects of macrophage mitochondrial transfer on cancer cells, we performed single cell RNA-sequencing on cancer cells that received macrophage mitochondria. These data revealed that mitochondrial transfer induced significant changes in canonical cell proliferation-related pathways (*Figure 1—figure supplement 2a*). To follow up on the RNA-sequencing results, we used flow cytometry to evaluate proliferation changes, and found significant increases in the percent of cells within the G2 and Mitotic (M) phases of the cell cycle in recipient cells, as compared to their co-cultured counterparts that did not receive mitochondria (*Figure 1e*; *Figure 1—figure supplement 2b-d*). These cells were not undergoing cell cycle arrest, as we found that recipient cells completed cytokinesis at rates equivalent to their co-cultured non-recipient counterparts (*Figure 1—figure supplement 2e*). For further confirmation of this proliferative phenotype, we used quantitative phase imaging (QPI) to detect changes in dry mass of co-cultured 231 cells over time (*Zangle and Teitell, 2014*). With this approach, we could obtain growth rate information of a large number of cancer cells over time (n=60 control cells; n=115 recipient cancer cells). Consistent with the flow cytometry-based cell cycle analysis, the specific growth rates increased significantly in 231 cells with macrophage mitochondria compared to 231 cells that did not receive mitochondria (*Figure 1f*). To examine whether the effects of mitochondrial transfer was sustained in recipient cells, we also measured the growth rates of daughter cells born from recipient 231 cells containing macrophage mitochondria (*Zangle et al., 2014*). We identified five 'parent' cancer cells with macrophage mitochondria, for which we were able to reliably follow both daughter cells upon division. Daughter cells that inherited the 'parent's' macrophage mitochondria exhibited an increase in their rate of change of dry mass over time versus sister cells that did not inherit macrophage mitochondria (*Figure 1—figure supplement 3a-c*). These experiments indicate that the proliferation phenotype in recipient cancer cells is sustained.

Our results so far suggest that either macrophage mitochondrial transfer increases cancer cell proliferation, or that more proliferative cells are simply more capable of receiving macrophage mitochondria. Thus, to test between these hypotheses, we first blocked cells in the G1-phase of the cell cycle by treating co-cultures with a CDK4/6 inhibitor, Palbociclib (*Figure 1—figure supplement 3d*), and we observed no changes in mitochondrial transfer rates (*Figure 1—figure supplement 3e*). These data indicate that the enhanced proliferation observed in recipient cells is not due to proliferative cells more readily receiving transfer.

We then performed experiments to rigorously test whether transferred macrophage mitochondria causes cancer cell proliferation, rather than mitochondrial receipt and proliferation being correlative events in cancer cells. We also wanted to determine whether the observed proliferative phenotype is due to macrophage mitochondria, and not other molecules that are passed along with the macrophage mitochondria. Thus, we biochemically purified mitochondria from a macrophage cell line, THP-1, and directly applied these macrophage mitochondria to cancer cells for 24 hr (*Figure 1g*). We then FACS-isolated cancer cell populations that contained purified macrophage mitochondria, and allowed this population to undergo additional rounds of cell division, and then reanalyzed the proliferative capacity of cancer cells that had retained the macrophage mitochondria versus cancer cells that had lost the macrophage mitochondria over this time. We first confirmed that cancer cells retained the macrophage mitochondria by imaging (*Figure 1h*). We also found that cancer cells that had retained the macrophage mitochondria exhibited an increased percentage of cells in the G2/M phase of the cell cycle compared to cancer cells that had lost the macrophage mitochondria (*Figure 1i-j*). Together with the QPI results, these results support the model that macrophage mitochondrial transfer promotes a sustained pro-growth and proliferative effect in both recipient and subsequent daughter cells.

## Transferred mitochondria are dysfunctional and accumulate ROS

We next sought to understand how donated mitochondria can stimulate a proliferative response in recipient cells. We performed time-lapse confocal microscopy on co-cultures and found that in cancer cells with macrophage mitochondria, macrophage-derived mito-mEm+ mitochondria remained distinct from the recipient host mitochondrial network. Cancer cells were cocultured with macrophages for 12 hr and subjected to an additional 15 hr of timelapse microscopy, and we observed no detectable loss of the fluorescent signal at transferred mitochondria throughout the course of imaging (*Figure 2a*, arrowhead; *Video 1*). Thus, transferred macrophage mitochondria did not appear to fuse with the existing endogenous mitochondrial network in recipient cells. To probe the functional state of the donated mitochondria, we performed live imaging with MitoTracker Deep Red (MTDR), a cell-permeable dye that is actively taken up by mitochondria with a membrane potential (*Poot et al., 1996*). To our surprise, all of the transferred mitochondria were MTDR-negative (*Figure 2b*, top left). This was also confirmed using a different mitochondrial membrane potential-sensitive dye, Tetramethylrhodamine Methyl Ester (TMRM; *Figure 1—figure supplement 1e*). These results suggested that the transferred mitochondria lacked membrane potential. To determine whether these membrane potential-deficient transferred mitochondria were subjected to lysosomal degradation, we labeled lysosomes and acidic vesicles with a dye, LysoTracker, and found that the majority of transferred mitochondria (57%) did not co-localize with the LysoTracker signal (*Figure 2b*, top right). The status of transferred mitochondria was unexpected because mitochondria typically maintain strong membrane potentials, and dysfunctional mitochondria that lack membrane potential are normally degraded or repaired by fusion with healthy mitochondrial networks (*Phinney et al., 2015*). Next, we utilized another dye which stains cellular membranes, MemBrite, and observed that 91% of transferred mitochondria were not encapsulated by a membranous structure, thus also excluding sequestration as a mechanism for explaining the lack of degradation or interaction with the endogenous mitochondrial network (*Figure 2—figure supplement 1a*). These data, taken together with the long-lived observation of the transferred mitochondria in *Figure 2a*, suggest that transferred macrophage mitochondria lack membrane potential, yet remain as a distinct population in recipient cancer cells, not fusing with the endogenous host mitochondrial network nor subjected to degradation.

Given the surprising observation that transferred mitochondria lack membrane potential, we hypothesized that instead of providing a metabolic or energetic advantage, the donated mitochondria may act as a signal source to promote sustained changes in cancer cell behavior. This hypothesis could offer insight into how this rare event, in which a relatively small amount of mitochondria is transferred,

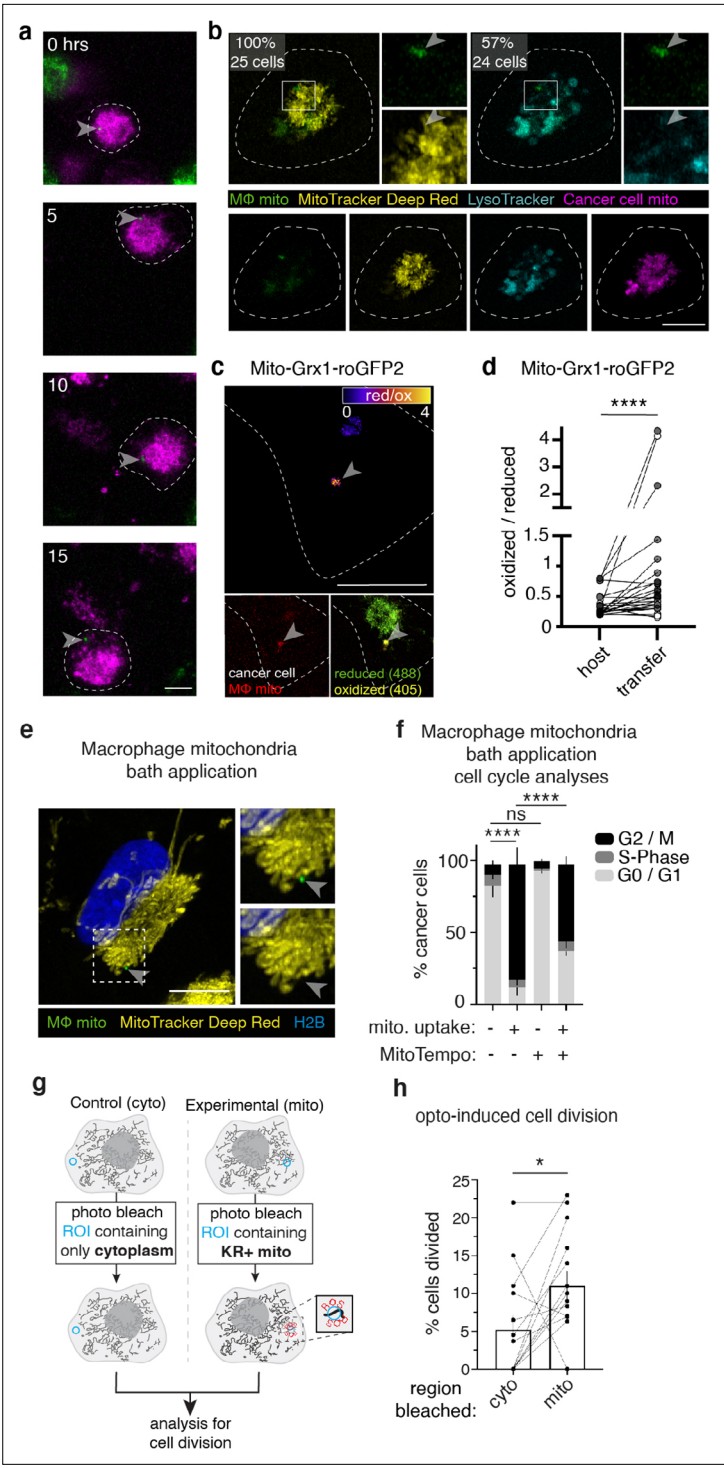

**Figure 2.** Transferred macrophage mitochondria are long-lived, depolarized, and accumulate reactive oxygen species, promoting cancer cell proliferation. (**a**) Stills from time-lapse imaging depicting the longevity of the transferred mitochondria (green, arrowhead) within a 231 cell (magenta, cell outline in white). Time elapsed listed in left corner. (**b**) Confocal image of a mito-RFP+ 231 cell (magenta) containing macrophage mitochondria (green, arrowhead) stained with MTDR (yellow) and LysoTracker (teal). MTDR does not accumulate in 100% of donated mitochondria (N=25 cells, 5 donors). Majority (57%) of donated mitochondria do not colocalize with LysoTracker signal (N=24 cells, 4 donors). (**c**) Ratiometric quantification of mito-Grx1-roGFP2 biosensor mapped onto the recipient 231 cell with fire LUT (top panel). Confocal image of mito-Grx1-roGFP2-expressing 231 cell (bottom right, green and yellow) containing a macrophage mitochondria (bottom left, red, arrowhead). (**d**) Ratiometric

*Figure 2 continued on next page*

*Figure 2 continued*

measurements of the mito-Grx1-roGFP2 sensor per 231 cell (paired dots) at a region of interest containing the host mitochondrial network (host) or a transferred mitochondria (transfer). Cells were co-cultured for 24 hr (N=27 cells, 3 donors indicated in shades of gray). (**e**) Exogenous purified macrophage mitochondria (green) is void of mitochondrial membrane potential (MitoTracker Deep Red-negative, yellow, arrowhead) in cancer cells. (**f**) Cell cycle analysis of cancer cells with exogenous purified macrophage mitochondria versus sister cells that did not take up exogenous purified mitochondria, either treated with vehicle or 100 μM mitoTEMPO (mitochondrially-targeted superoxide scavenger. N=3 donors; statistics for G2/M only). (**g**) Schematic of optogenetic experiments to generate data in (**h**). Cells expressing mito-KillerRed are photobleached in a specific ROI containing either cytoplasm only (left) or mito-KillerRed+ mitochondria (right). Following photobleaching, cells are imaged over time to quantify the amount of cell division. (**h**) Quantification of cell division after photobleaching. Each data point is the average within a field of view (N=13 experiments), with control (cyto) and experimental (mito) data shown as paired dots per experiment. Scale bars are 10 μm. Wilcoxon matched-pairs signed rank test (**d, h**), two-way ANOVA (**f**), *p<0.05; ****p<0.0001.

The online version of this article includes the following figure supplement(s) for figure 2:

**Figure supplement 1.** Transferred mitochondria accumulate reactive oxygen species, and internalized exogenous mitochondria are not encapsulated in a membrane compartment.

**Figure supplement 2.** Inducing reactive oxygen species results in cancer cell proliferation.

could mediate sustained changes in the proliferative capacity of recipient cancer cells. One signaling molecule associated with mitochondria is reactive oxygen species (ROS), which occur normally as byproducts of mitochondrial respiration, and can be produced at high levels during organellar dysfunction (*Schieber and Chandel, 2014*). Using a genetically encoded biosensor, mito-Grx1-roGFP2, as a live readout of the mitochondrial glutathione redox state (*Gutscher et al., 2008*), we found that after 24 and 48 hr, significantly higher ratios of oxidized:reduced protein were associated with the transferred mitochondria versus the host network (*Figure 2c–d*; *Figure 2—figure supplement 1b*). These data indicate that transferred macrophage mitochondria in recipient cells are associated with higher levels of oxidized glutathione, suggesting that they are accumulating higher amounts of ROS. Consistent with these results, a second biosensor that is specific for the reactive oxygen species $H_2O_2$, mito-roGFP2-Orp1 (*Gutscher et al., 2009*), also reported more oxidation at the transferred mitochondria compared to the host network (*Figure 2—figure supplement 1c–d*) after 48 hr of co-culture. At 24 hr, we observed a similar trend, but no statistically significant difference (*Figure 2—figure supplement 1d*). These results indicate that ROS accumulate at the site of transferred mitochondria in recipient cancer cells. It is unclear whether the observed ROS accumulation is generated by the transferred mitochondria themselves, or generated elsewhere in the recipient cancer cell and accumulating locally at transferred mitochondria. Regardless of the source, we observed robust ROS accumulation specifically at the site of transferred mitochondria and with this unexpected finding, we next tested whether this ROS accumulation could serve as a molecular signal, regulating cell proliferation.

To rigorously test the model that transferred macrophage mitochondria accumulate ROS, promoting cancer cell proliferation, we turned toward purified macrophage mitochondria approaches as in *Figure 1g* and sought

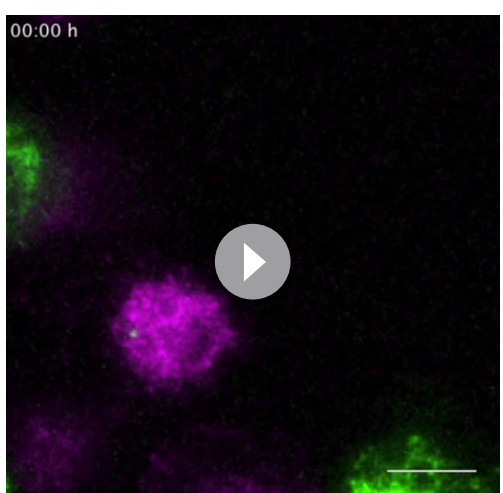

00:00 h

**Video 1.** Macrophage mitochondria are long-lived and remain distinct in recipient cancer cells. Video depicting a recipient mito-RFP expressing 231 cell (magenta) that contains mito-mEm macrophage mitochondria (green in magenta cell, center of frame). 231 cells were co-cultured with macrophages for 7 hr prior to the start of imaging for a duration ~15 hr with a time interval of 5 min. Maximum intensity projections of images are displayed at 12 frames per second, timestamp in upper left corner in hours (h), and scale bar is 10 μm.

https://elifesciences.org/articles/85494/figures#video1

approaches to reduce ROS levels. First, to better model the macrophage mitochondrial transfer to cancer cells that occurs in coculture conditions, we determined conditions for cancer cells to internalize exogenous macrophage mitochondria at rates similar to in vitro mitochondrial transfer conditions at 24 hr – 0.68% ± 0.36% internalization rate, n=3 biological replicates (compare to *Figure 1d*). We next determined that purified mitochondria taken up by cancer cells remain distinct, are not encapsulated by membranes after 24 hr (*Figure 2—figure supplement 1e*), and do not exhibit membrane potential (*Figure 2e*). Similar to our previous proliferation results with purified macrophage mitochondrial uptake at longer time points (*Figure 1j*), we found that cancer cells with internalized purified macrophage mitochondria (which, under these conditions, comprise ~1% of the total population) exhibited a significant increase in proliferative cells in the G2/M phase of the cell cycle, compared to sister cells that did not internalize mitochondria (*Figure 2f*, comparing black bars in lanes 1&2), and that this increase was ameliorated when ROS is quenched with a mitochondrially localized superoxide scavenger, mitoTEMPO (*Figure 2f*; comparing black bars in lanes 2&4). Importantly, cancer cells that did not internalize mitochondria were not affected by ROS quenching (*Figure 2f*; comparing black bars in lanes 1&3). These results indicate that transferred mitochondria promote proliferation in a ROS-dependent manner.

To test whether ROS accumulation can induce cancer cell proliferation directly, we stably expressed a mitochondrially localized photosensitizer, mito-KillerRed, which generates ROS when photobleached with 547 nm light (*Bulina et al., 2006*). As expected, photobleaching mito-KillerRed+ regions of interest induced ROS (*Bass et al., 1983*; *Figure 2—figure supplement 2a*). We then drew mito-KillerRed+ regions of interest that mimicked the size of macrophage mitochondrial transfer to induce local ROS in cancer cells, and analyzed the rate of cell division by imaging these cells over 18 hr (*Figure 2g*). We found that cells with induced ROS (by photobleaching mito-KillerRed+ regions) exhibited an increased percentage of dividing cells compared to negative control photobleached cells (mito vs. cyto bleach; *Figure 2h*; *Figure 2—figure supplement 2b–c*). These results indicate that induction of mitochondrially localized ROS can directly promote cancer cell proliferation.

## ROS accumulation leads to ERK-dependent proliferation

We next aimed to determine how ROS induction may regulate cell proliferation. ROS is known to induce several downstream signaling pathways (*Schieber and Chandel, 2014*; *Brillo et al., 2021*), including ERK/MAPK signaling, a pathway known to regulate proliferation and tumorigenesis (*Dhillon et al., 2007*). Thus, we sought to determine if cancer cells that had received macrophage mitochondria exhibited increased ERK signaling. We stably expressed the ERK-Kinase Translocation Reporter (ERK-KTR) (*Regot et al., 2014*), which translocates from the nucleus to the cytoplasm when ERK is activated, in 231 cells (231-ERK-KTR). After co-culturing 231-ERK-KTR cells with macrophages, we used the imaging flow cytometer, Amnis ImageStream, to compare relative ERK-KTR translocation values in hundreds of cells that had or had not received macrophage mitochondria (ERK-KTR quantification and ERK signaling validation described in *Figure 3—figure supplements 1–2*). These data show that cancer cells with macrophage mitochondria have significantly higher cytoplasmic to nuclear (C/N) ERK-KTR ratios compared to cells that did not receive mitochondrial transfer, indicating increased ERK activity (*Figure 3a–b*; *Figure 3—figure supplement 2a–b*).

Due to our observations that cells that receive macrophage mitochondria exhibit increased ERK activation and that local ROS induction is sufficient to induce cell proliferation, we then asked whether cancer cell mitochondrial ROS would directly enhance ERK activation. By expressing both mito-KillerRed and ERK-KTR in 231 cells, we induced ROS by photobleaching mito-KillerRed+ regions and found that ROS induction increased ERK-KTR translocation, indicating that ROS induction is sufficient to increase ERK activity in cancer cells (*Figure 3c–d*; *Figure 3—figure supplement 2c*). We next tested whether ERK signaling is required for the mitochondrial transfer-induced cancer cell proliferation. We first determined an effective concentration of SCH772984, an ERK inhibitor (ERKi), that still inhibits ERK activity, but does not dramatically affect 231 proliferation, as we sought to determine whether inhibiting ERK affects mitochondrial transfer-induced proliferation, not proliferation more generally. We first confirmed that treatment with this effective concentration of ERKi led to decreased ERK activity, as determined by the ERK-KTR translocation reporter (*Figure 3—figure supplement 4a–b*). We then found that treatment with ERKi significantly decreased proliferation of recipient 231 cells when compared to vehicle control-treated recipient cells (*Figure 3e*; *Figure 3—figure supplement*

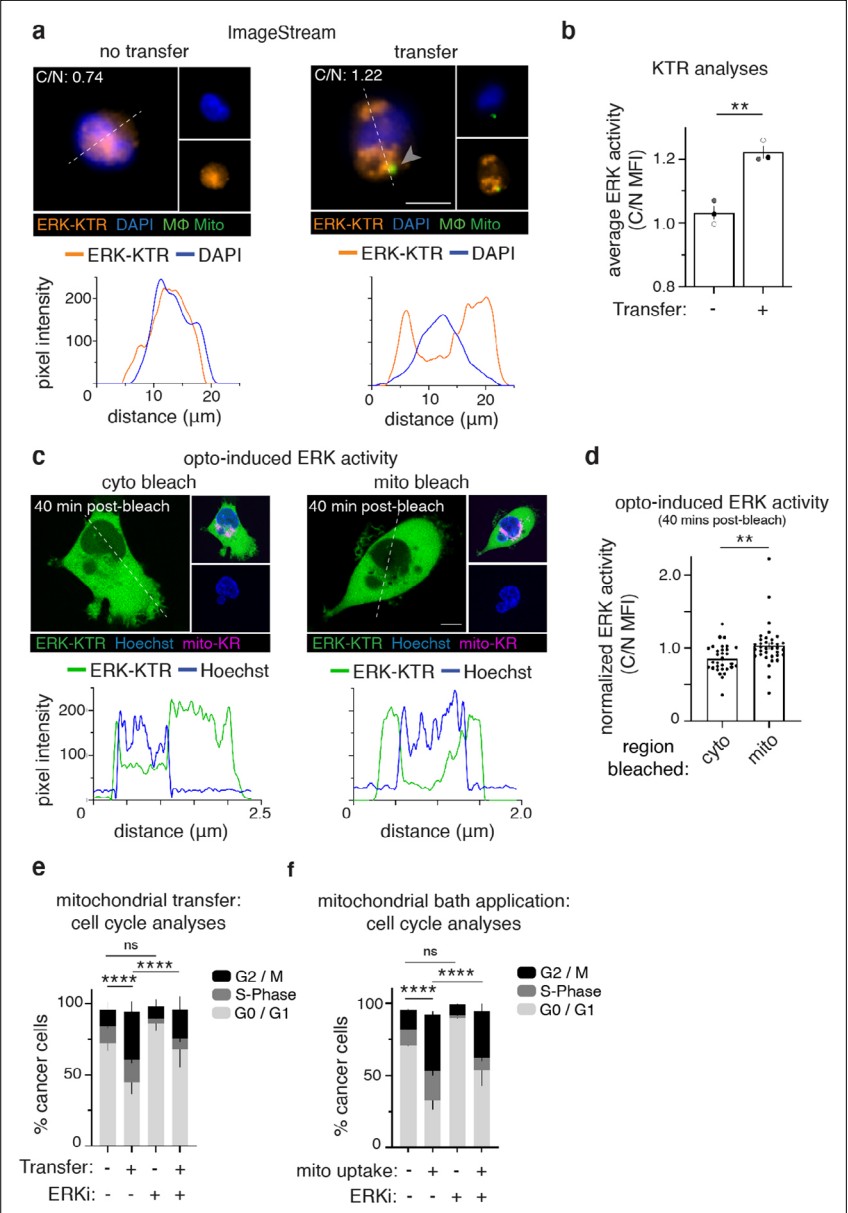

**Figure 3.** Recipient cancer cells exhibit ERK-dependent proliferation. (**a**) ImageStream was used to measure the MFI of an ERK-Kinase Translocation Reporter (ERK-KTR, orange) in the nucleus (DAPI, blue) or cytoplasm of co-cultured 231 cells that did (right) or did not (left) receive mitochondria (green, arrowhead). Below: representative line scans (white dotted lines) of ERK-KTR (orange) and DAPI (blue). (**b**) Average ERK activity from data displayed in (**d**) (cytoplasm/nucleus (C/N) mean fluorescence intensity (MFI); N=3 donors indicated as shades of gray). (**c**) Confocal images of 231 cells expressing ERK-KTR (green) and Mito-KillerRed (magenta) with Hoechst 33342 (blue), after control cytoplasmic bleach (cyto, left) or mito-KillerRed+ bleach (mito, right). Below: representative line scans (white dotted lines) of ERK-KTR (green) and Hoechst (blue). (**d**) Quantification of ERK-KTR translocation 40 min post-bleach (cyto vs. mito), normalized to time 0. Each dot represents a measurement from a single cell. (**e**) Analysis of proliferative capacity by quantifying Ki-67 and DNA levels of co-cultured 231 cells treated with vehicle or ERK inhibitor (ERKi) with or without transfer or (**f**), mitochondrial internalization after mitochondrial bath application (N=3 donors; statistics for G2/M only). Error bars represent SEM and scale bars are 10 μm., Welch's t-test (**b**), Mann-Whitney (**d**), two-way ANOVA (**e–f**), *p<0.05; **p<0.01; ****p<0.0001.

The online version of this article includes the following figure supplement(s) for figure 3:

**Figure supplement 1.** Amnis ImageStream pipeline for ERK-KTR quantification.

**Figure supplement 2.** ERK-KTR analysis and validation using the Amnis ImageStream pipeline.

*Figure 3 continued on next page*

**Figure supplement 3.** Quantification of ERK activity in recipient 231 cells or upon ROS induction.

**Figure supplement 4.** ERK inhibition reduces proliferation in cancer cells with macrophage mitochondria.

*4c*). We further noted that the decrease in proliferation with this concentration of ERK inhibitor was observed only in cancer cells that received macrophage mitochondria, and not in cancer cells that did not receive macrophage mitochondria (bars 2&4 in *Figure 3e*, compared to bars 1&3), suggesting that the ERK-dependent cell proliferation specifically occurs in cancer cells that received mitochondrial transfer. As a control, we also confirmed that ERKi treatment did not alter mitochondrial transfer efficiencies, showing that ERK signaling does not influence mitochondrial transfer (*Figure 3—figure supplement 4d*). Finally, similarly to *Figure 2f*, we bath applied purified macrophage mitochondria to cancer cells in the presence of vehicle or ERKi and compared the proliferative capacity of cells that had internalized macrophage mitochondria versus cells that did not (*Figure 3f*). We found that, as before, uptake of purified macrophage mitochondria increased the percentage of cancer cells in the G2/M phase of the cell cycles (*Figure 3f*, bars 1&2), but that this process is ameliorated by the inhibition of ERK signaling (*Figure 3f*, bars 2&4). We also found that ERK inhibition did not affect the cell cycle state of cancer cells that had not taken up purified macrophage mitochondria (*Figure 3f*, bars 1&3). Thus, these results indicate that mitochondrial transfer promotes cancer cells proliferation through a ROS/ERK-dependent mechanism.

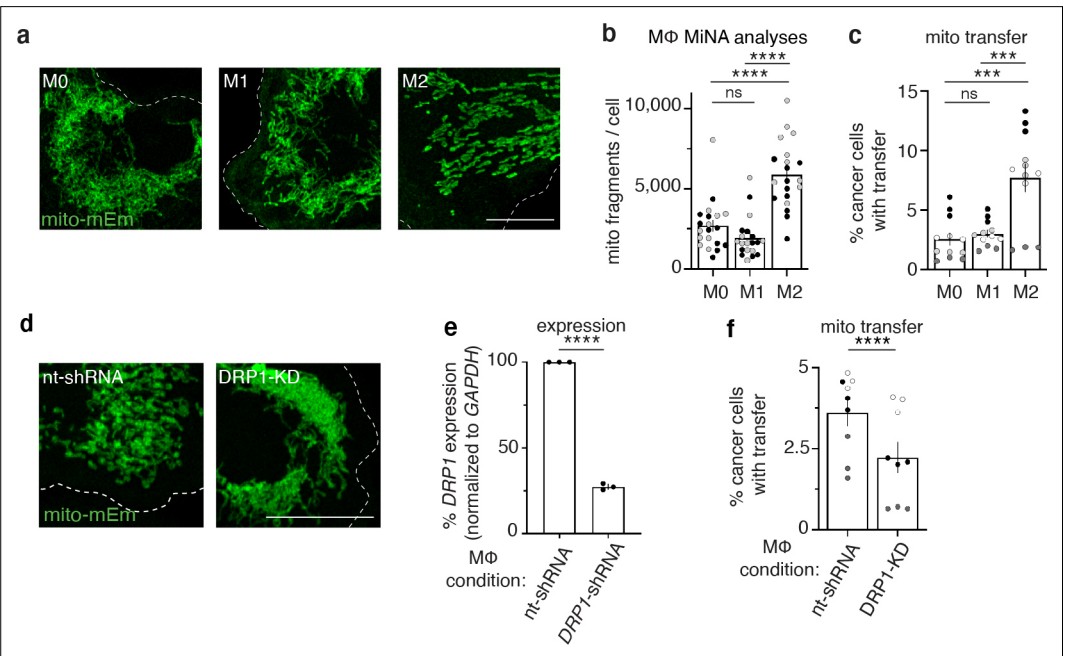

**Figure 4.** M2-like macrophages exhibit increased mitochondrial fragmentation and increased mitochondrial transfer to cancer cells. (**a**) Representative images of mito-mEm+ macrophages that were non-stimulated (M0, left) or activated to become M1-like (middle) or M2-like (right). (**b**) Mitochondrial network analyses (MiNA) were used to determine number of mitochondrial fragments per cell (N=2 donors). (**c**) Macrophages were co-cultured with mito-RFP 231 cells for 24 hr and mitochondrial transfer was quantified with flow cytometry (N=4 donors). (**d**) Representative images of mito-mEm (green) macrophages in macrophages with control nt-shRNA KD and DRP1 KD. (**e**) q-RT-PCR of DRP1 knockdown (DRP1-KD) macrophages (N=3 donors). (**f**) Rates of mitochondrial transfer with control and DRP1-knockdown macrophages (N=3 donors). For all panels, individual donors are indicated as shades of gray with each cell as a data point, error bars represent SEM and scale bars are 10 µm. Two-way ANOVA (**b, c**), unpaired t-test (**e, f**), ***p<0.001; ****p<0.0001.

The online version of this article includes the following figure supplement(s) for figure 4:

**Figure supplement 1.** M2-like macrophages exhibit increased mitochondrial transfer to cancer cells.

**Figure supplement 2.** Macrophages transfer mitochondria to breast cancer patient-derived cells.

## M2-like macrophages exhibits enhanced mitochondrial transfer rates

In many solid tumors, it has long been appreciated that macrophage density is associated with disease progression and poor patient prognosis (*Pollard, 2004*). Macrophages are highly plastic, altering their phenotypes, expression profiles and function, depending on environmental stimuli and conditional requirements (*Pan et al., 2020*). Accordingly, macrophages exist in a spectrum of activation states but are canonically simplified by the two ends of spectrum: pro-inflammatory and anti-tumorigenic M1-like macrophages; or anti-inflammatory and pro-tumorigenic M2-like macrophages (*Huang et al., 2018*). Since the ways in which M2-like macrophages promote tumor progression continue to be elucidated and given that there remains a dearth of understanding of how donor cell biology affects mitochondrial transfer, we aimed to determine how macrophage activation status affects intracellular mitochondrial dynamics and transfer efficiencies to cancer cells.

Activated macrophages were co-cultured with 231 cells, and we first quantified mitochondrial networks using Mitochondrial Network Analyses (MiNA) (*Valente et al., 2017*). We found that M2-like macrophages contain significantly more fragmented mitochondria when compared to M1-like or M0 (non-activated) macrophages (*Figure 4a–b*; *Figure 4—figure supplement 1a–d*). We then co-cultured 231 cells with either M0, M1-like, or M2-like macrophages and using flow cytometry, we found that mitochondrial transfer efficiencies were significantly increased from M2-like macrophages when compared to M1-like or non-activated M0 macrophages (*Figure 4c*). Given that M2-like macrophages exhibited fragmented mitochondrial networks and enhanced mitochondrial transfer rates, we hypothesized that smaller mitochondrial fragments might be transferred more readily than larger networks. To test this hypothesis, we directly manipulated mitochondrial morphology by modulating a key regulator of mitochondrial fission, DRP1 (*Fonseca et al., 2019*). Macrophages transduced with *DRP1*-shRNA containing lentivirus exhibited hyper-fused mitochondrial networks (*Figure 4d and e*) and exhibited decreased mitochondrial transfer (*Figure 4f*). Together these findings reveal that macrophage activation alters mitochondrial dynamics, and that altering mitochondrial dynamics directly affects mitochondrial transfer rates. Finally, to determine whether the functionality of transferred mitochondria differ between macrophage subtypes, we evaluated the membrane potential of transferred mitochondria, and found that transferred mitochondria from M1-like and M2-like macrophages were similarly depolarized (*Figure 4—figure supplement 1e*), as to what we observed with M0 macrophages (*Figure 2b*). Taken together, these results suggest that pro-tumorigenic M2-like macrophages exhibit increased mitochondrial fragmentation, promoting mitochondrial transfer to cancer cells.

To assess whether mitochondrial transfer also occurs in a clinically relevant cancer model, we used three-dimensional stable organoid cultures generated from patient-derived xenografts (PDxOs) (*Guillen et al., 2021*). We examined organoids from a recurrent primary breast tumor (HCI-037) and a bone metastasis (HCI-038) derived from the same breast cancer patient. PDxOs grown in 3D (*Figure 4—figure supplement 2*, top) were dissociated, combined with mito-mEm+ macrophages (*Figure 4—figure supplement 2a*, bottom), and then embedded in Matrigel (experimental scheme in *Figure 4—figure supplement 2b*). After 72 hr, mitochondrial transfer was assayed by live imaging (*Figure 4—figure supplement 2c*) and quantified with flow cytometry (*Figure 4—figure supplement 2d, e*, ). Mitochondrial transfer was observed from macrophages to both HCI-037 and HCI-038 PDxO cells (*Figure 4—figure supplement 2e*), although intriguingly, M2-like macrophages preferentially transferred mitochondria to the bone metastasis PDxO cells (HCI-038), whereas M0 and M2-like macrophages transferred mitochondria to primary breast tumor PDxO cells (HCI-037) at the same rate. In all cases, transferred macrophage mitochondria lacked membrane potential (*Figure 4—figure supplement 2c*), consistent with our results in 231 recipient cells.

## Cancer cells with macrophage mitochondria exhibit increased proliferation in vivo

Next, to better model a tumor environment, we examined macrophage mitochondrial transfer to cancer cells in two separate in vivo models of metastatic breast cancer. We first injected E0771 murine adenocarcinoma cells expressing mito-mEm into wildtype C57BL/6J mice that had received lethal irradiation with subsequent bone marrow reconstitution from mito::mKate2 mice (mito:mKate2→WT), restricting mKate2 expression to immune cells (experimental schematic in *Figure 5—figure supplement 1a*). We found that in vivo mitochondrial transfer occurred at a rate of 4.8%, compared to control transplantation studies at 0.46% (*Figure 5a*). We also performed experiments in mice that

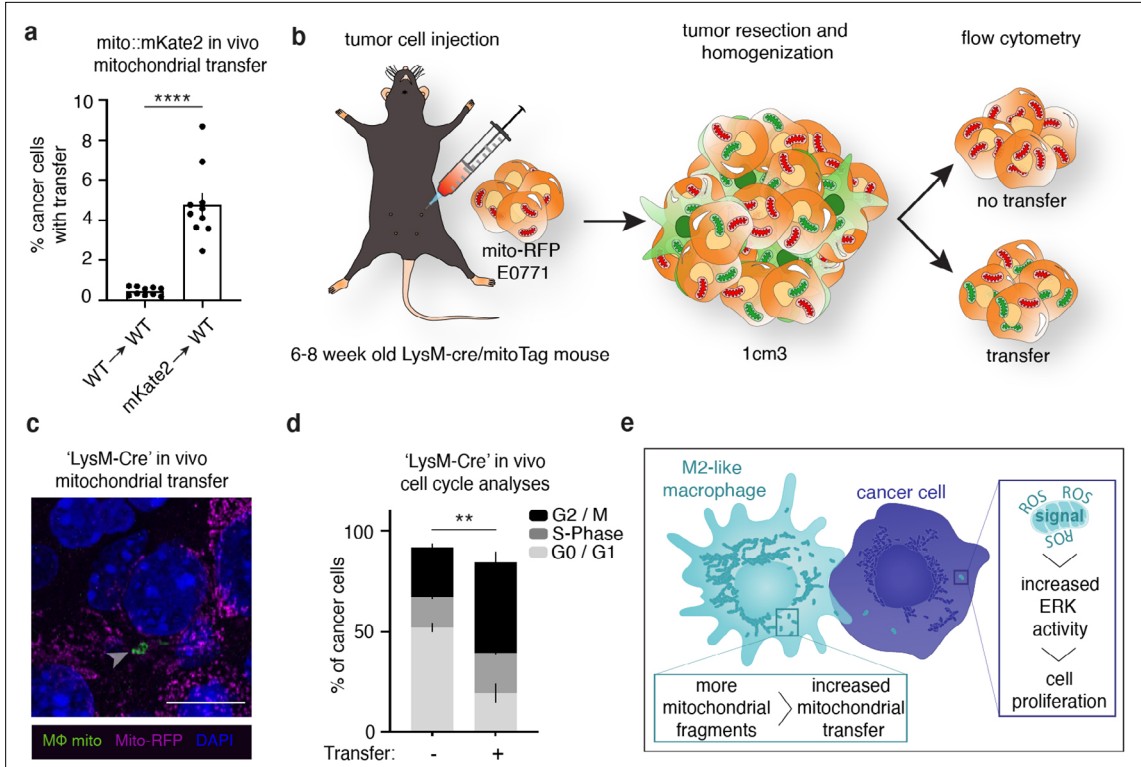

**Figure 5.** Macrophage mitochondrial transfer promotes tumor cell proliferation in vivo. (**a**) Quantification of E0771 mammary adenocarcinoma cells from in vivo tumors with mKate2+ mitochondria in bone marrow reconstitution experiments versus control mice. N=10 mice per condition. (**b**) Schematic representation of a second mouse model to quantify proliferation in cancer cells with macrophage mitochondria in vivo. Myeloid lineages were specifically labeled with mito-GFP by crossing a Loxp-Stop-Loxp-MitoTag-GFP mouse to a LysM-Cre mouse. E0771 cells expressing mito-RFP were injected into the mammary fat pad of mice with MitoTag-GFP expression in myeloid cells, and tumors were isolated and analyzed for direct observation of transfer through fluorescent microscopy (**c**) and Ki67/DNA to quantify proliferative index (**d**). (**c**) Representative immunofluorescence image of E0771 tumor cell expressing mito-RFP (magenta) containing GFP+ macrophage mitochondria (arrowheads) from mice in which GFP+ mitochondria are restricted to the myeloid lineage ('LysM-Cre'). (**d**) Cell cycle analysis of E0771 in vivo tumor cells with and without GFP+ macrophage mitochondria in 'LysM-Cre' model in which GFP+ mitochondria are restricted to the myeloid lineage. N=3 mice. (**e**) Working model for macrophage mitochondrial transfer to breast cancer cells. For all panels, individual donors are indicated as shades of gray with each cell as a data point, error bars represent SEM and scale bars are 10 μm. Welch's t- test (**a**), two-way ANOVA (**d**), **p<0.01; ****p<0.0001.

The online version of this article includes the following figure supplement(s) for figure 5:

**Figure supplement 1.** Murine mammary adenocarcinoma cells with macrophage mitochondria exhibit increased cell proliferation in vivo.

**Figure supplement 2.** Predicted increase in population size due to transferred mitochondria as a function of number of population doublings.

restrict GFP-labeled macrophage mitochondria to the myeloid lineage by using the LysM-Cre transgenic mouse crossed to the lox-stop-lox-MitoTag mouse (experimental schematic *Figure 5b*, see methods for more details). We injected E0771 cells expressing mito-RFP into these mice with GFP-labeled macrophage mitochondria and observed E0771 cells containing macrophage mitochondria using immunohistochemistry approaches of tumor sections (*Figure 5c*; *Figure 5—figure supplement 1b*). Using similar cell proliferation analyses as previously described (*Figure 1e*), we also observed that recipient tumor cells exhibited enhanced proliferative capacity compared to the tumor cells that did not receive transfer (*Figure 5d*; *Figure 5—figure supplement 1c*). These results show that mammary adenocarcinoma cells with macrophage mitochondria exhibit increased proliferation in vivo.

Finally, to determine the impact of macrophage mitochondrial transfer on population growth over time, we derived a relationship between overall growth of the cell population and the fraction of cells with macrophage mitochondria that experience an increase in growth rate (*Figure 5—figure supplement 2*). We used this analysis to predict the increase in population size due to transferred mitochondria as a function of the number of population doublings. The fraction of the population receiving mitochondria was assumed to be 5% based on our in vivo studies (*Figure 5a*), with the tumor cell

population exhibiting a 15% increase in baseline growth rate due to transferred mitochondria based on QPI growth rate measurements (*Figure 1f*). We also assumed that half of the population loses transferred mitochondria, and the associated growth increase, with every division given that our QPI measurements indicated that typically only one of the daughter cells inherit the parent's exogenous macrophage mitochondria (*Figure 1—figure supplement 3a–c*). By 20 divisions, with a 15% increase in growth rate (brown line), even with only 5% of the cancer cell population with macrophage mitochondria at any given time, the model already predicts a 15% increase in population size compared to baseline population rates (comparing the brown line to the blue dotted line). These results highlight the significance of macrophage mitochondrial transfer on the growth of a cell population over time. Taken together, our work supports a model (*Figure 5f*) whereby M2-like macrophages exhibit fragmented mitochondria leading to increased mitochondrial transfer. In the recipient cancer cell, transferred mitochondria are long-lived, depolarized, and accumulate ROS, leading to increased ERK activity and subsequent cancer cell proliferation.

## Discussion

Lateral mitochondrial transfer is a relatively young and rapidly evolving field. Previously literature had shown that healthy mitochondria are transferred, enhancing recipient cell viability by increasing ATP production and stimulating metabolic processes. Our observations, however, suggest that transferred mitochondria promote tumor cell proliferation as a byproduct of their potential dysfunctionality. This model raises several fascinating questions, including when and how transferred mitochondria become depolarized and accumulate ROS, where the ROS is generated in the recipient cell, and why depolarized mitochondria are not repaired or degraded in the recipient cell, given that 231 cells are capable of performing mitophagy (*Biel and Rao, 2018*). Impaired mitophagy and enhanced mitochondrial dysfunction are hallmarks of age (*Chen et al., 2020*), yet little is known about how age-related mitochondrial dysfunction influences mitochondrial transfer. Interestingly, instead of degrading dysfunctional mitochondria through mitophagy, neurons in an Alzheimer's disease mouse model have been shown to transfer dysfunctional mitochondria to neighboring astrocytes (*Lampinen et al., 2022*), which contributes to neuronal mitochondrial homeostasis. Given that age is the greatest known risk factor of Alzheimer's disease, and most cancers are also age-related, these data collectively warrant broader investigations into how age-associated mitochondrial dysfunction contributes to mitochondrial transfer and how this form of communication may have specific influences on distinct diseased states.

Cellular stress occurs throughout biological systems, and cells have evolved a myriad of mechanisms to cope with disadvantageous cellular conditions, including mitochondrial stress (*Ma et al., 2020*). Our results suggest that transferred mitochondria are a source for downstream signal activation through a ROS-ERK-mediated mechanism. The origin of ROS generation in recipient cells is still unclear, and how ROS locally accumulate at macrophage mitochondria is an open question. It is possible that ROS are generated by the transferred mitochondria themselves, as previous reports have shown that mitochondria with reduced membrane potential can generate ROS (*Feng et al., 2022*; *Franco-Iborra et al., 2018*; *Nakai et al., 2003*). However, there are multiple mechanisms for ROS production (*Zhao et al., 2019*), and it is possible that the ROS are generated elsewhere in the cell and accumulating at transferred mitochondria. A previous report showed that the endoplasmic reticulum can produce ROS in the presence of dysfunctional mitochondria (*Leadsham et al., 2013*), suggesting another possible explanation. The questions of how ROS is generated, and how ROS can be spatially restricted to a specific subcompartment of the cell are exciting avenues of investigation and much further study. Although high levels of ROS are cytotoxic to cells (*Stadtman and Levine, 2000*; *Fruhwirth and Hermetter, 2008*; *Auten et al., 2002*), physiological levels of ROS are known second messenger molecules stimulating various pro-survival signaling cascades (*Schieber and Chandel, 2014*; *Brillo et al., 2021*). Additionally, a modest increase of mitochondrial-derived ROS has been shown to exhibit protective mechanisms through mitohormesis (*Crewe et al., 2021*; *Ristow and Schmeisser, 2014*), a process in which cellular defense mechanisms are stimulated by sub-lethal stress levels, protecting cells to withstand a secondary exposure. Mitochondrial transfer has been shown to promote resistance to subsequent chemotherapeutic treatments in healthy neurons (*English et al., 2020*) and tumor cells (*Wang et al., 2018*; *Boukelmoune et al., 2018*), however the mechanism of this resistance is unclear. It is possible that mitochondrial transfer mediates this protective

response through mitohormesis, promoting longevity and proliferation of the recipient cancer cells. More studies are required to connect mitochondrial transfer, sub-lethal cellular stress, and resistance to chemotherapeutic treatments in disease progression and tissue homeostasis.

The role of mitochondrial transfer has been largely studied in recipient cells. There remains a dearth of information describing how donor cells regulate mitochondrial transfer. Although intercellular mitochondrial transport has been implicated in the process of mitochondrial transfer (*Boukelmoune et al., 2018*; *Ahmad et al., 2014*), we show that macrophage differentiation directly affects mitochondrial transfer through changes in their mitochondrial morphology. The relationship between macrophage differentiation and metabolism has been partially defined, with anti-tumor-like (M1-like) macrophages exhibiting more glycolytic metabolism, and pro-tumor-like (M2-like) macrophages upregulating oxidative phosphorylation (*Van den Bossche et al., 2017*; *Mortezaee and Majidpoor, 2022*). But how mitochondrial morphology regulates metabolism is less understood. Studies have indicated that mitochondrial fusion supports increases oxidative phosphorylation in fibroblasts (*Yao et al., 2019*), however other studies have shown that increasing mitochondrial fission upregulates oxidative phosphorylation in hepatocytes (*Zhou et al., 2022*). Thus, the correlation between mitochondrial morphology and cellular metabolic status is unclear, and these differences are likely due to different cell types and environmental conditions. While how the metabolic status of donor cells influences mitochondria is still unknown, our results support the hypothesis that pro-tumorigenic M2-like macrophage activation promotes mitochondrial fragmentation, and that mitochondrial fragmentation directly promotes mitochondrial transfer.

Our findings are consistent with several studies describing a metastatic advantage in cancer cells that receive exogenous mitochondria (*Zampieri et al., 2021*; *van der Merwe et al., 2021*; *Dong et al., 2017*; *Tan et al., 2015*). However, the mechanism underlying this behavior is unexpected. Studies examining mitochondrial transfer have typically used recipient cells with damaged or non-functional mitochondria, and the fate and function of donated mitochondria are rarely followed in recipient cells. Furthermore, it was largely unclear how transferred mitochondria can affect the behavior of recipient cells with functioning endogenous mitochondrial networks, particularly if the donated mitochondria only account for a small fraction of the total mitochondrial network in the recipient cell. Our work detailing how transferred mitochondria can activate downstream signaling pathways in response to ROS provides an explanation for how a relatively small amount of transferred mitochondria can generate a sustained behavioral response in recipient cells.

## Materials and methods
### Cell culture of cell lines and peripheral blood mononuclear cells (PBMCs)

Human cell lines MDA-MB-231 (HTB-26), MDA-MB-468 (HTB-132), A375 (CRL-1619), THP-1 (TIB-202), MCF10A (CRL-10317), and the murine cell lines E0771 (CRL-3461) were directly purchased from American Type Culture Collection and cultured according to their recommendations. Cell lines are authenticated through STR profiling, and all cultured cell lines are subjected to mycoplasma testing every 6 months using the Universal Mycoplasma Detection Kit (30–1012 K, ATCC). Base medias used were DMEM, high glucose (11965118, ThermoFisher), RPMI (11875119, ThermoFisher) and 10% heat-inactivated fetal bovine serum (FBS; F4135, ThermoFisher). All cell lines were kept in culture for no more than 25 passages total.

### Genetic modification of PBMCs and differentiation into macrophages

PBMCs were isolated from leukocyte filters obtained from de-identified human blood donors (ARUP Blood Services). CD14 +monocytes were isolated from buffy coats and genetically modified with lentiviral vectors in the presence of virus-like particles packaging Vpx (to overcome restriction in myeloid cells) as previously described (*Johnson et al., 2020*; *Greiner et al., 2022*). Briefly, freshly harvested CD14 +monocytes were plated at a density of 4–5 M cells per 10 cm plate in 'macrophage culture media' containing: RPMI (11875119, ThermoFisher), 10% FBS (26140079, Thermo Fisher), 0.5% penicillin/streptomycin (P/S; P4333, Thermo Fisher), 10 mM HEPES (15630080, ThermoFisher), 0.1% 2-Mercaptoethanol (21985023, Thermo Fisher), recombinant human GM-CSF at 20 ng/ml (300–03, Peprotech) with the addition of polybrene (1 µg/ml), and supernatant containing Vpx particles (0.5 mL

per 4 M cells) to facilitate viral transduction. Thirty min after plating, 100–200 µL of concentrated lentiviral stock was added to the plated monocytes. 50% of the media was replaced on day 2 and a full media replacement occurred on day 4. Macrophages were used for experiments starting on day 6 or 7 after harvest of PBMCs unless otherwise noted.

## Distinction of biological and technical replicates

Each human blood donor (referred to as 'donors' or 'experiments') is a biological replicate. Multiple samples from each donors run in parallel are defined as technical replicates (typically in triplicate for each biological replicate).

## Generation of mito-FP and FP-TOMM20 stable cell lines

We generated a modified pLKO.1 plasmid backbone with an accessible multiple cloning site (pLKO.1_MCS) for generation of fluorescent reporters. For mito-FP expression, we cloned the cytochrome oxidase subunit VIII mitochondrial targeting sequence and tagged it to *mEmerald* (referred to as mito-mEm) or *tagRFPt* (referred to as mito-RFP) and introduced these into the pLKO.1_MCS backbone in order to generate lentiviruses. pLKO.1 mito-mEmerald and pLKO.1 mito-TagRFP-T are available on Addgene (#174542 and 174543, respectively). For FP-TOMM20 expression, inserts containing the sequence of either *mEmerald* (mEmerald-TOMM20) or *mcherry* (mCherry-TOMM20) fused to *TOMM20* and cloned into the pLKO.1 backbone and used to generate lentiviruses. Stable lines were generated through lentiviral transduction. For transduction, approximately 50,000 cells were plated into one well of a 6-well plate directly into the appropriate lentivirus supernatant diluted 1:5 in DMEM complete media with a final concentration of 10 µg/mL polybrene (TR-1003-G, Sigma). After 48–72 hr, cells were expanded, and multiclonal populations were flow sorted for appropriate levels of fluorescent expression. All other transgenic cell lines were generated as outlined in subsequent sections.

mEmerald-TOMM20-N-10 (Addgene plasmid # 54282) and mCherry-TOMM20-N-10 (Addgene plasmid # 55146) were a gift from Michael Davidson.

## Lentivirus production

pLKO.1_MCS plasmids containing the appropriate transgene were used to generate lentivirus as outlined in *Johnson et al., 2020*. Briefly, 293 FT cells in 15 cm plates were transfected with PEI-max (24765, Polysciences) and plasmids for pCMV-VSV-G, psPax2, and transgene cassettes. The following day, cells were washed and cells were grown for an additional 36 hr in fresh media. Supernatants were harvested, passed through 0.45 µm filters, and either used fresh or concentrated by ultracentrifugation as previously described (*Johnson et al., 2020*). Lentiviral supernatants were used to transduce cell lines as outlined in 'generation of mito-FP' section unless otherwise noted.

## Flow cytometry

The following flow cytometry machines were used: a BD FACS Aria (equipped with 4 Lasers: 405, 488, 561, 640) referred to as Aria, or a BD LSR Fortessa (5 Lasers: UV, 405, 488, 561, 640) referred to as the Fortessa. Technical details per experiment type are listed below.

## Stable line generation

Cells were enzymatically dissociated using trypsin-EDTA (25200056, ThermoFisher) and resuspended in buffer consisting of 0.5% Bovine Serum Albumin (BSA; Sigma, A9418) in DPBS (14190250, ThermoFisher). Cells were sorted according to fluorescent intensity on the Aria and collected in the appropriate media containing 0.5% P/S.

## Mitochondrial transfer quantification

For MDA-MB-231 and MCF10A cell lines: cells were enzymatically dissociated using trypsin-EDTA and stained as follows: cells were resuspended in 'staining buffer' (DPBS + 2% FBS) containing a human antibody against CD11b conjugated to the fluorophore Brilliant Violet 711 (BV711-CD11b; macrophage marker; Biolegend, 301344) at a 1:20–40 dilution. After a 30 min incubation on ice, cells were washed and resuspended in cold DPBS for analysis on the Fortessa. The background level of mEmerald fluorescence was set at 0.2% based on a fully stained co-culture control where macrophages were not transduced with mito-mEmerald. This gate was defined by FACS-isolating co-cultures of mito-RFP

MDA-MB-231/mito-mEm macrophages and determining a gate that accurately isolated MDA-MB-231 cells containing macrophage mitochondria. We found that the cancer cell population with the highest mEm signal were cancer/macrophage fusions, and we therefore removed this population from downstream analysis. Setting the gate to 0.2% predominantly led to isolation of cancer cells with fragments of macrophage mitochondria, as visualized by microscopy.

## Mitochondrial transfer quantification of PDxO containing co-cultures

Hanging drop co-cultures suspended in Growth Factor Reduced Matrigel (354230, Corning) were pooled and dissociated using a solution of Dispase II (50 U/mL; 17105041, Fisher Scientific) followed by TrypLE Express (12605010, Thermo Fisher). Cells were then incubated in TrueStain FcX (422301, ThermoFisher) at 1:33 dilution with staining buffer for 10 min at room temperature. Primary human antibodies against CD326 conjugated to PE (PE-EpCam; PDxO marker; 369806, Biolegend) and BV711-CD11b were added at 1:20 and 1:40, respectively. After 30 min on ice, cells were washed and resuspended in cold DPBS for analysis on the Fortessa. The background level of mEmerald fluorescence in the 'transfer gate' was set at 0.2% based on a fully stained monoculture control.

## Quantification of Ki67 and DNA content

Co-cultures were enzymatically dissociated with trypsin-EDTA and incubated in staining buffer containing anti-human BV711-CD11b at 1:40 for 30 min on ice. Cells were then fixed and stained using the eBioscience Foxp3/Transcription Factor Staining Buffer Set (00-5523-00, ThermoFisher) according to manufacturer's instructions. Cell were stained with an APC conjugated Ki67 antibody (APC-Ki67; 17-5699-42, ThermoFisher) at 1:20 for 30 min followed by a 3 µM DAPI (D9542, Sigma) solution for 10 min. Cells were resuspended in cold DPBS for analysis on the Fortessa. The background level of mEmerald fluorescence in the 'transfer gate' was set at 0–0.2% based on a fully stained co-culture control where macrophages were not transduced with mito-mEmerald.

## Single-cell RNA-seq

Mito-mEm macrophages were cocultured with mito-RFP 231 cells for 24 hr. Two populations were FACS-isolated: (1) Mito-RFP 231 cells containing mito-mEm macrophage mitochondria; (2) mito-RFP 231 cells not containing mito-mEm macrophage mitochondria. From FACS-isolated populations, a cDNA library was generated using the 10 X genomics Single Cell 3' Gene Expression Library V3 and amplified according to the manufacturer's protocol. The resulting libraries were sequenced on a NovaSeq 6000 resulting in approximately 100 K mean reads per cell. The raw sequencing data were processed using CellRanger 3.02 (https://support.10xgenomics.com/) to generate FASTQ files, aligned to GRCh38 (Ensemble 93), and a gene expression matrix for individual cells based on the unique molecular indices was generated. The resultant filtered gene-cell barcode matrix was imported into SEURAT version 4 (*Hao et al., 2020*) with R studio version 1.3.1093 and R version 4.03. We first performed quality control by determining the mean and standard deviation of genes per cell and filtered out all cells that were more than 1.5 standard deviations away from the mean. The reads were then scaled and normalized using SEURAT 'sctransform' function (*Hafemeister and Satija, 2019*). Using the normalized data, we determined differential gene expression in the MDA-MB-231 population that received macrophage mitochondria compared to those that did not, using a non-parametric Wilcoxon rank sum test with the SEURAT 'FindMarkers' function. Lastly, the differential expression data were exported from R and pathway enrichment analysis was performed using Qiagen's Ingenuity Pathway Analysis software (*Krämer et al., 2014*). Single-cell RNA-sequencing data are available with GEO accession number GSE181410. The analysis code for single-cell RNA-sequencing analysis is available on GitHub (https://github.com/rohjohnson-lab/kidwell_casalini_2021; RRID:SCR_002630 (version number 1)).

## Trans-well experiments

Approximately 40,000 mito-RFP MDA-MB-231 and 80,000 mito-mEm macrophages were plated in trans-wells (3401, Corning) under the conditions listed in *Figure 1—figure supplement 1e–h*. Cells were analyzed after 24 hr with flow cytometry as indicated in 'mitochondrial transfer quantification' section.

## Live cell imaging of co-cultures with cell-permeable dyes

Imaging was performed using either a Zeiss LSM 880 with AiryScan technology (Carl Zeiss, Germany) and a 63 x/1.4 NA oil objective or a Leica Yokogawa CSU-W1 spinning disc confocal microscope with a Leica Plan-Apochromat 63 x/1.4 NA oil objective and iXon Life 888 EMCCD camera. Images taken on the LSM 880 were acquired using the AiryScan Fast mode. For all live imaging, cells were maintained at 37 °C, 5% $CO_2$ with an on-stage incubator.

MDA-MB-231 cells and primary macrophages stably expressing the appropriate transgenes were mixed in a 1:2 ratio and plated at an approximate density of 300,000 cells directly onto 35 mm glass bottom dishes (FD35-100, World Precision Instruments) for all live imaging experiments unless otherwise noted. Duration of co-culture is indicated in main text or figure legend.

For detection of nuclear and mitochondrial DNA, Hoechst 33342 (B2261, Sigma) was diluted into the culture media to a final concentration of 5 µg/mL. After 10 min at 37 °C, cells were washed, and complete media was replaced before imaging.

For detection of mitochondrial membrane potential with MitoTracker Deep Red (MTDR; M22426, ThermoFisher), MTDR was diluted into serum-free DMEM media (11965118, ThermoFisher) at a final concentration of 25 nM and incubated at 37 °C for 30 min. Following incubation, cells were washed with warm PBS, and warmed complete media was replaced before imaging.

For detection of mitochondrial membrane potential with Tetramethylrhodamine, Methyl Ester, Perchlorate (TMRM; T668, ThermoFisher), TMRM was diluted into serum-free DMEM media at a final concentration of 100 nM and incubated at 37 °C for 30 min. Following incubation, cells were washed with warm PBS, and warmed complete media was replaced before imaging.

For detection of lysosomes and acidic vesicles, LysoTracker Blue (L7525, ThermoFisher) was diluted to a final concentration of 75 nM in serum-free DMEM media and incubated at 37 °C for 30 min. Following incubation, cells were washed with warm PBS and warmed complete media was replaced before imaging.

For detection of plasma/vesicular membranes, MemBrite 640/660 (Biotium, 30097) was used at a final concentration of 1:1000 and stained according to manufacturer's instructions. To preferentially label intracellular membrane compartments, cells were allowed to rest for 45 min after Membrite 640/660 staining before imaging, as indicated in the manufacturer's instructions.

For detection of ROS, Carboxy-$H_2$DCFDA (C400, ThermoFisher) was diluted to 5 µM into warmed HBSS (14025092, ThermoFisher) and incubated at 37 °C for 15–30 min. After incubation, cells were washed with HBSS and warmed complete media was replaced before imaging.

## Live imaging of sorted recipient cells

MDA-MB-231 cells were harvested and stained as indicated in 'mitochondrial transfer' section of flow cytometry methods. Cells were sorted on the Aria directly into media containing 0.5% P/S. Sorted cells were plated directly onto imaging dishes coated with CellTak (354240, Corning) and allowed to attach at 37 °C for up to 4 hr before staining and live imaging.

## Quantitative phase imaging (QPI)

Mito-RFP MDA-MB-231 cells and mito-mEm macrophages were seeded in a 1:2 ratio at a density between 90,000 and 120,000 cells directly onto imaging dishes 24 hr prior to the start of imaging. QPI images were acquired on Olympus IX83 inverted microscope (Olympus Corporation, Japan) with Phasics SID4 camera (Phasics, France) and Thorlabs 623 nm wavelength DC2200 LED (Thorlabs, USA). The microscope was operated in brightfield with Olympus UPLFLN 40 X objective and a 1.2 X magnifier in front of camera, giving ×48 magnification. Fluorescence images were acquired using X-Cite 120LED illumination (Excelitas technologies, USA) and an R1 Retiga camera (Cairn research Ltd, UK) with GFP (Olympus Corporation U-FBNA) and RFP (IDEX health & science, USA mCherry-B-000) filter cubes. Cells were maintained at 37 °C temperature, 5% $CO_2$ and 90% humidity with an Okolab (Okolab, Italy) on-stage incubator on a Prior III Proscan microscope stage (Prior Scientific Instruments Ltd., UK). Automation was performed with MicroManager open-source microscopy software via MATLAB 2012b. QPI images of 40 positions per imaging set, four replicate (biological replicate) imaging sets total, were acquired every 15 min with fluorescence images acquired in an alternate subset of locations every 15 min for 48 hr to reduce phototoxicity.

## QPI data analysis

QPI and fluorescent images were analyzed with MATLAB 2019a. Cell phase shift images were background corrected using sixth order polynomial surface fitting, and converted to dry mass ($m$) map, using, $m = \int \frac{1}{\alpha} \phi \lambda dA$, where $\lambda$, is the wavelength of source light = 0.623 µm, $\alpha$, specific refractive increment = 0.185 µm$^3$/pg, $A$, image pixel area = 0.36 µm$^2$/pixel, and $\phi$ is the phase shift in fraction of a wavelength at each pixel. Cell dry mass maps were then segmented using a Sobel filter for edge detection and tracked over time (*Crocker and Grier, 1996*). Specific growth rate of each tracked cell was computed as the slope of a linear, least-squares best fit line to mass over time data normalized by cell average mass. Fluorescent mitochondria images were resized to match QPI images and overlaid with corresponding QPI image segmentation mask to measure the integrated fluorescence intensity of every cell, normalized by cell area. Macrophage mitochondria high frequency punctae signal in MDA-MB-2321 cells were separated from the high intensity, low spatial frequency of the macrophage mitochondria network fluorescence signal using the rolling ball filter in MATLAB. The size of the rolling ball was 4.8–9 µm, chosen to be just above the average size of mitochondrial punctae based on the quantity of mitochondria transferred and retained in the MDA-MB-231 cells. Cells with RFP signal 1.5 times more than the background were identified as MDA-MB-231 cells, and with mEmerald fluorescent signal double that of background as macrophages. MDA-MB-231 cell tracks were then binned based on the presence or absence of mEmerald +mitochondrial punctae, indicating transfer from macrophages. The specific growth rate of each cell was calculated as the slope of a least-squares linear fit to QPI mass vs time data divided by the average mass of the cell. The code for automated tracking of cell mass from QPI and fluorescence data and computing growth rates of the different groups of cells is available on GitHub (https://github.com/Zangle-Lab/Macrophage_tumor_mito_transfer, copy archived at *ZangleLab, 2023*).

## Cytokinesis analysis

Cytokinesis rate was calculated by tracking cells manually to confirm division of cells in less than the maximum doubling time expected (40 hours). Cells leaving the imaging frame in less than 30 hr were omitted from the cytokinesis calculation.

## Lineage analysis

The average specific growth rate of MDA-MB-231 parent and daughter cell was calculated by manually annotating mass versus time tracks from mass tracking based on the presence of mEmerald +punctae. The difference in growth of daughter cells that did or did not inherit mitochondria from mitochondria containing parents was observed by normalizing the mass of each daughter by its initial mass at birth.

## ROS biosensor line generation, imaging, and quantification

MDA-MB-231 cells were transfected with the following plasmids: pLPCX mito-Grx1-roGFP2 (*Gutscher et al., 2008*) and pLPCX mito-roGFP2-Orp1 (*Gutscher et al., 2009*) (Addgene, plasmid #64977 and #64992, respectively) using the Polyplus-transfection jetPRIME DNA/siRNA transfection kit (55–131, Genesee Scientific) according to the manufacturer's instructions. Cells were allowed to recover for 3–7 days and then sorted for expression. Cells were passaged every 3–5 days and sorted as needed to maintain a high percentage of expressing cells. Biosensor-expressing MDA-MB-231 lines were co-cultured with mito-RFP expressing macrophages for 24 or 48 hr and imaged on the Zeiss LSM 880. Cells were sequentially imaged (per z-plane) for the presence of transferred macrophage mitochondria (Ex. 561 nm, Em. BP 570–620nm +LP 645 nm) and the biosensor in its a reduced (Ex. 488 nm, Em. BP 420–480nm +BP 495–550 nm) and oxidized (Ex. 405 nm, Em. BP 420–480nm +BP 495–550 nm) form. Images were initially processed using Zen software (see image analysis section) and further analysis was performed using FIJI as indicated in *Morgan et al., 2011*. pLPCX mito Grx1-roGFP2 (Addgene plasmid # 64977) and pLPCX mito roGFP2-Orp1 (Addgene plasmid #64992) were a gift from Tobias Dick.

## Mito-KillerRed line generation and imaging

To generate *3xHA-killerred-OMP25*, a plasmid containing *3xHA-EGFP-OMP25* (*Chen et al., 2016*) was used as a template and the sequence of KillerRed replaced EGFP. The entire transgene was then cloned into the pLKO.1_MCS backbone. pLKO.1 3xHA-KillerRed-OMP25 is available on Addgene

(#174544). MDA-MB-231 cells were transduced with lentiviral supernant that packaged the *3xHA-killerred-OMP25* transgene, allowed to recover and were cell sorted to select for the appropriate level of fluorescent expression. Cells expressing both mito-mEm and mito-KillerRed were generated in parallel to confirm the correct localization of the mito-KillerRed (data not shown).

pMXs-3XHA-EGFP-OMP25 was a gift from David Sabatini (Addgene plasmid #83356).

### For generation of mt-ROS with the mito-KillerRed cell line

MDA-MB-231 mito-KillerRed-expressing cells were labeled with Carboxy-H$_2$DCFDA as described above. Using a Leica Yokogawa CSU-W1 spinning disc confocal microscope equipped with a 2D-VisiFRAP Galvo System Multi-Point FRAP/Photoactivation module, MDA-MB-231 mito-KillerRed-expressing cells were imaged at 488 nm (for DCFDA detection) and 561 nm (for mito-KillerRed detection) at a time interval of 2 seconds. After 2 frames, a~2µm x 2µm region of interest (ROI) of mito-KillerRed was photobleached using a 561 laser (100% laser power, 5ms, 1 cycle), and continuous imaging at 488 nm and 561 nm allowed for DCFDA quantification and mito-KillerRed photobleaching, respectively.

### To quantify cell division upon ROS production

Cells were stained with 5 µg/mL Hoescht 33342 as described above to visualize nuclei. Multiple stage positions were established such that control experiments, in which a cytoplasmic ROI without mito-KillerRed expression that was photobleached using identical parameters, as well as a no-photobleaching control, could be imaged simultaneously with experimental photobleached cells. Approximately 8–10 cells of each category – photobleached in mito-KillerRed-expressing regions, photobleached in control cytoplasmic non-expressing regions, or not photobleached – were imaged by acquiring Z-stacks (1 µm step size) every 15 min for 18 hr. Cell division was quantified by visualizing nuclear division with FIJI software.

### ERK-KTR generation

MDA-MB-231 cells were transduced with lentiviral supernant that was packaged using either pLentiPGK Blast DEST ERKKTRmRuby2 or pLentiPGK Puro DEST ERKKTRClover plasmids (*Kudo et al., 2018*) as outlined in 'generation of mito-FP' section. Cells were allowed to recover post-infection, sorted for fluorescent expression, and maintained as stable cell lines.

pLentiPGK Blast DEST ERKKTRmRuby2 (Addgene plasmid # 90231) and pLentiPGK Puro DEST ERKKTRClover (Addgene plasmid # 90227) were a gift from Markus Covert.

### ERK-KTR-mClover and mito-KillerRed generation and imaging

A stable MDA-MB-231 line expressing mito-KillerRed was transduced with lentivirus that was packaged using a pLentiPGK Puro DEST ERKKTRClover plasmid. Cells were sorted for expression of both mito-KillerRed and ERK-KTR-mClover and maintained as a stable line.

### For mt-ROS generation and ERK-KTR imaging

MDA-MB-231 cells expressing mito-KillerRed and ERK-KTR-mClover were stained with 5 µg/mL Hoescht 33342 as described above to visualize nuclei. Using a Leica Yokogawa CSU-W1 spinning disc confocal microscope equipped with a 2D-VisiFRAP Galvo System Multi-Point FRAP/Photoactivation module, MDA-MB-231 cells expressing mito-KillerRed and ERK-KTR-mClover were imaged every 1 minute with 561 nm (for mito-KillerRed) and 488 nm (for ERK-KTR-mClover) and 405 nm (for nuclei) lasers. A~2 µm x 2 µm ROI of KillerRed + mitochondria was photobleached using a 561 laser (100% laser power, 5ms, 1 cycle), and continuous imaging at 488 nm and 561 nm allowed for visualization of ERK-KTR-mClover translocation and mito-KillerRed photobleaching, respectively. Multiple stage positions were set such that control experiments, in which a cytoplasmic region without mito-KillerRed expression that was photobleached using identical parameters, could be imaged simultaneously with experimental photobleached cells.

### ERK-KTR quantification with FIJI

ERK-KTR-mClover translocation was quantified every 10 minutes by taking maximum projections of Z-planes only encompassing the cell nucleus. Using FIJI software, a ROI was drawn in the nucleus

guided by the Hoescht staining, and the MFI of ERK-KTR-mClover was quantified in this region. The same ROI was moved outside of the nucleus to a cytoplasmic region devoid of mitochondria, and the MFI of ERK-KTR-mClover was quantified. This analysis was performed for each timepoint after photobleaching. The values were then used to calculate a cytoplasmic:nuclear ratio at each time point, and normalized to 1 at time point zero.

## Quantification of ERK-KTR using the Amnis ImageStream

To quantify translocation of the ERK-KTR-mRuby we used the Amnis Imagestream mk II with ISX software (version 201.1.0.725). Mito-mEm macrophages were co-cultured with ERK-KTR-mRuby+MDA MB-231 cells for 24 hr. Samples were prepared as indicated in 'quantification of Ki67 and DNA content' section with the exception that we did not stain for intracellular markers. Images were captured with the 40 x objective and sample collect flow was set to low, as this allows for higher image resolution. Using Image Data Exploration and Analyses Software (IDEAS; version 6), we quantified translocation using two metrics: (1) the IDEAS translocation Wizard and (2) custom-generated program to detect cytoplasmic (cyto) and nuclear (nuc) ERK-KTR mean fluorescent intensities (MFI) to calculate a cyto:nuc ratio as indicated in *Figure 3—figure supplements 1 and 2*. The translocation wizard is a pre-built program made to detect the nuclear translocation of a probe. It does this by making a pixel-by-pixel correlation between the probe of interest (ERK-KTR) and the nuclear image (DAPI). The program gives each cell a score indicating how similar the two fluorescent images are. A high score suggests the images are similar (more nuclear translocation) and a low score suggests that the images are less similar (less nuclear translocation). We also quantified ERK-KTR translocation by generating a custom masking strategy to quantify the mRuby MFI in the cytoplasm and nucleus using IDEAS software. To identify nuclear mRuby, we manually set a threshold of DAPI signal and reported the mRuby MFI of pixels within that threshold range. To quantify the cytoplasmic mRuby fraction we reported the mRuby MFI from outside the threshold. These values are then used to calculate a cyto:nuc ratio.

## Drug treatments: ERKi, PMA and MitoTEMPO

SCH772984 (ERKi; 7101, SelleckChem) and Phorbol 12-myristate 13-acetate (PMA; S7791, Selleck-chem) was dissolved in 100% DMSO to make 10 mM stock solutions and stored at –80 °C. No individual aliquot went through more than 2 freeze-thaw cycles. The stock solution was thawed and then diluted directly into complete media for a final concentration of 1 µM for ERKi and 100 nM (cancer cell treatment) or 162 nM (THP-1 differentiation) for PMA. For all ERKi experiments, co-cultures were treated at the time of plating and for a duration of 24 hr. For PMA treatment of cancer cells the cells were plated the day prior and were treated for 1 hr prior to harvest and analysis. THP-1 cells were differentiated for 24 hr in PMA prior to harvest. To quench mitochondrial ROS, MitoTEMPO (Cayman Chemical, 16621) was formulated at 200 mM in 100% DMSO and diluted directly into warm complete media for a final concentration of 100 µM. MDA-MB-468 cells were treated for the duration of 24 hours as described in 'Mitochondrial isolation and bath application', and cells were harvested for proliferative capacity analyses as previously in 'Quantification of Ki67 and DNA content'. MitoTEMPO aliquots were stored at –20 °C, remained protected from light and never underwent a freeze-thaw cycle.

## Macrophage activation and verification

For macrophage activation, macrophages were harvested and differentiated as indicated in 'cell culture of PBMCs' section. Between days 6–7 of differentiation, IFN-γ (3000–02, Peprotech, 20 ng/mL) for M1 activation or IL-4 +IL-13 (200–04, 200–13, Peprotech, 20 ng/mL) for M2 activation were added to culture media for 48 hr before experiments were conducted. To confirm M1 and M2 activation, macrophages were collected and stained for known surface markers for M1 (CD86; 62-0869-42, Thermofisher) and M2 (CD206; 321110, Biolegend) activation. Flow cytometry was performed on the Fortessa to observe changes in fluorescent intensities across M0, M1, and M2 macrophages populations (*Figure 4—figure supplement 1a*).

## Immunofluorescence and analysis of mitochondrial morphology

Mito-mEm expressing macrophages were co-cultured with mito-RFP +MDA MB-231 cells for 24 hr and fixed with warm 4% PFA with 5% sucrose in 1 x DPBS for 20 min and permeabilized with 0.2%

Triton X-100 in 1 x DPBS (9002-93-1, Sigma). Cells were stained with chicken α-GFP (AB13970, Abcam) and Rabbit α-RFP (AB62341, Abcam) antibodies at 1:500 and 1:1000, respectively. The following secondary antibodies were used: Alexa Fluor 488 AffiniPure Goat anti-Chicken (103-545-155, Jackson ImmunoResearch) and IgG (H+L) Cross-Adsorbed Goat anti-Rabbit Alexa Fluor 555 (A21428, Invitrogen) both at 1:500. Cells were subsequently stained with 1 µg/mL DAPI in DPBS for 10 min. Cells were then mounted with ProLong Diamond Antifade Mountant (P36965, ThermoFisher) and stored at 4 °C before imaging. Imaging was performed using the Zeiss LSM 880 using the AiryScan fast mode. AiryScan processed images (see image analysis section) were used to quantify mitochondrial morphologies with the FIJI plug-in, Mitochondrial Network Analyses (MiNA; *Figure 4—figure supplement 1b–d*). Pre-processing parameters: Manually select top and bottom of the cell of interest, exclude any space above and below the cell as this can introduce background noise. 3D project cell. Unmask sharp Radius (5), Mask Weight (0.6), Median 3D (0.5, 0.5, 0.5), Make binary (Otsu), Skeletonize, Analyze skeleton 2D/3D. A 'mitochondrial fragment' was defined as a mitochondrion with 0–1 branches, 0 junctions, and a length greater than 0 µm and a maximum length of 2 µm.

## DRP1 knockdown

Monocytes were isolated as indicated in 'cell culture of PBMCs' section and transduced with lentiviruses to express mito-mEm and either non-target (nt) short hairpin (sh) RNA (SHC002, Sigma), or *DRP1*-shRNA (TRCN0000001097, Sigma; gene target HGNC ID 2973). All constructs were either produced or cloned into the pLKO.1 backbone.

## rt-qPCR verification of genetic knockdown

RNA from nt-shRNA and *DRP1*-shRNA expressing macrophages were isolated from 3 independent macrophage donors. To isolate RNA, we used standard TRIzol/chloroform RNA isolation techniques. cDNA libraries were made using SuperScript III Reverse Transcriptase (18080093, ThermoFisher), according to manufacturer's instructions. *DRP1*-knockdown was verified via qRT-PCR with Power SYBR Green Mast Mix (4368511, ThermoFisher). Primers were designed with NCBI primer design, commercially produced by Integrated DNA Technologies and tested for specificity with standard PCR. Primers were as follows; *DRP1*-F: AGAAAATGGGGTGGAAGCAGA, *DRP1*-R: AAGTGCCTCTGA TGTTGCCA, *GAPDH*-F: AGCCACATCGCTCAGACA, *GAPDH*-R: ACATGTAAACCATGTAGTTGAGGT . Cycle Thresholds (CT) values were determined by averaging 3 technical replicates from 3 biological samples. Control ΔCT: expression was normalized to *GAPDH* by subtracting the *DRP1* CT value of the nt-shRNA expressing macrophages from the *GAPDH* CT value of the same sample. Target gene, *DRP1*ΔCT: *DRP1* CT values of the *DRP1*-shRNA expressing macrophages were subtracted from the *GAPDH* CT values of the same sample. The ΔΔCT values was calculated by subtracting *DRP1*ΔCT – control ΔCT. Normalized target gene expression was calculated ($2^{-\Delta\Delta CT}$) and used to determine % knockdown (($1–2^{-\Delta\Delta CT}$)*100).

## PDxO culture and co-culture with macrophages

PDxO cell lines HCI-037 and HCI-038 were generated and maintained as described in *Guillen et al., 2021*. Like MDA-MB-231 cells, these models are estrogen and progesterone receptor negative and HER2 negative (triple negative breast cancer). For co-culture with macrophages, mature PDxOs were dissociated from Growth Factor Reduced Matrigel with a Dispase II solution followed by treatment with TrypLE Express to generate a suspension of single cells. PDxO cells were then mixed with mito-mEm macrophages (differentiated for 7–9 days) in a 1:2 ratio at a density of 90,000 cells total per hanging drop culture. Macrophage media was used for hanging drops (for media components, see isolation of PBMCs section) and they were suspended from the lid of a tissue culture plate to allow for cell aggregation for 24 hr and then pooled and embedded into Growth Factor Reduced Matrigel. Embedded hanging drop cultures were then allowed to incubate for 72 hr and were then analyzed for mitochondrial transfer with flow cytometry (see flow cytometry section).

## Mitochondrial isolation and bath application

For data represented in *Figure 1g–j*: 150–200x10^6 mito-mEm expressing THP-1 cells were pelleted by centrifugation for 5 min at 300 *g*. Pellets were resuspended in 2 mL of mitochondrial isolation buffer (70 mM sucrose, 220 mM D-mannitol, 2 mM HEPES, 1 x protease inhibitor, pH 7.4) and incubated on

ice for 15–30 min. Suspended cells were dounce homogenized 100–150 times in a Potter-Elvehjem PTFE pestle and glass tube (Sigma, P7734). Cell homogenates were centrifuged at least twice (700 g for 10 min, 4 °C) to pellet and remove unwanted cellular material, until no pellet was observable – as many as 7 centrifugation cycles. Final supernatants were centrifuged at 20,000 g for 15 min at 4 °C to pellet isolated mitochondria. Mitochondrial pellets were suspended in ~250 µL of ice cold mitochondrial isolation buffer +protease inhibitor, and relative mitochondrial concentrations were determined via standard BCA protein concentration assay (ThermoFisher, 23225). 20–30 µg/mL of mitochondria were applied to pre-plated mito-RFP expressing MDA-MB-231 cells for 18–24 hours. After mitochondrial incubation cells were thoroughly washed to remove any un-internalized mitochondria and mitochondrial percent internalization was determined via flow cytometry and cells were FACS isolated with BD FACS Aria as described above under 'Flow cytometry – Stable line generation'. FACS isolated cells were either imaged on the LSM880 Airy Scan Confocal as described in 'Live cell imaging of co-cultures with cell-permeable dyes', or plated for an additional 48 hr. After roughly two cell cycles, the cells were harvested and cell cycle analyses were conducted as described in 'Quantification of Ki67 and DNA content'.

For data represented in *Figures 2e–f , and 3f*: Mito-mEm expressing THP-1 monocytes were differentiated with 162 nM Phorbol 12-myristate 13-acetate (PMA - SelleckChem, #S7791) for 24 hr. Cells were trypsinized and washed with ice cold PBS and centrifuged at 300 *g* for 5 min. Cell pellets were suspended in 500–1000 µL mitochondrial isolation buffer +protease inhibitor and dounce homogenized as reported above. Final supernatants were centrifuged at 20,000 *g* for 15 min at 4 °C to pellet isolated mitochondria. Mitochondrial pellets were suspended in 110 µL of ice cold mitochondrial isolation buffer +protease inhibitor, and mitochondrial concentrations were determined via standard BCA (as above). Pre-plated MDA-MB-468 cells were bath applied with concentrations 3–5 µg/mL of exogenous mitochondria for 5–6 hr which was then removed to eliminate any un-internalized mitochondria. Twenty-four hr after initial mitochondrial addition, cells were either imaged on the LSM880 Airy Scan Confocal as described in 'Live cell imaging of co-cultures with cell-permeable dyes', treated with ERK inhibitor as described in 'Drug treatments: ERKi, PMA and MitoTEMPO' and harvested for cell cycle analyses were conducted as described in 'Quantification of Ki67 and DNA content'. All drug treatments (ERK inhibitor and MitoTEMPO) were applied at the time of mitochondrial application and were maintained until harvest.

## In vivo models

All animal experiments were approved by the Institutional Animal Care and Use Committee (IACUC) at the University of Utah (protocol # 19–12001) and at the Cleveland Clinic (protocol #2179). In accordance to approved protocols, all animals were anesthetized appropriately to assure maximum comfort throughout the duration of procedures. When tumors were grown to approved volumes, mice were humanely euthanized with slow $CO_2$ gas exchange for 5 min. We calculated how many animals would be required for each experiment using G*Power3.1 – Based on our in vitro studies, we considered a 5% increase in mitochondrial transfer or a 5% increase in the percentage of cells in the G2/M phase of the cell cycle as statistically significant, with a 1% standard deviation, thus, we required a minimum of three animals per treatment group. With the variability in tumor growth, we injected at least five animals per treatment group such that we could ensure to complete studies with at least three animals. Regarding *Figure 5a*: Six-week-old C57BL/6J (The Jackson Laboratory, Stock #000664) and Tg(CAG-mKate2)1Poche/J (The Jackson Laboratory, mito::mKate2, stock #032188) female mice were purchased from the Jackson Laboratory as required and housed in the Cleveland Clinic Biological Research Unit Facility. Wild-type mice were treated with 11 Gy radiation split into two fractions. $2\times10^6$ bone marrow cells from mito:mKate2 or wild-type mice were retro-orbitally injected for reconstitution. Drinking water was supplemented with Sulfatrim (Pharmaceutical Associates, Inc) during the first 10 days, and mice were monitored for an additional 6 weeks. A total of 250,000 mito-mEm E0771 cells were mixed with 1:50 diluted Geltrex (ThermoFisher) and implanted to 4th mammary pad in 100 µl RPMI. Mice were treated with Buprenorphine and Ibuprofen for 3 days, and monitored for endpoint symptoms. Animals were euthanized when the tumors reached 1 cm³ or 10% of the body weight was lost. Resected tumors were minced and incubated with Collagenase IV (StemCell Technologies) containing DNAseI (Roche) for 30 min at 37 °C. Single cells were strained through 70 µm filter (FisherBrand) and stained with 1:1000 diluted LIVE/DEAD Fixable Stains (ThermoFisher) for 10 min on ice. Samples were acquired with BD Fortessa.

For in vivo cell cycle analysis upon macrophage mitochondrial transfer in *Figure 5d*: 8–12 week old B6N.Cg-Gt(ROSA)26Sortm1(CAG-EGFP*)Thm/J (also known as MitoTag mice, The Jackson Laboratory stock 032675 *Fecher et al., 2019* and B6.129P2-Lyz2tm1(cre)Ifo/J (also known as LysMcre mice, The Jackson Laboratory, stock 004781) were ordered and crossed accordingly, producing offspring which were heterozygous for both transgenes. These heterozygous siblings were crossed to produce both experimental (MitoTag/cre) and control (WT/cre) animals in the same litter. 250,000 Mito-RFP E0071 cells were injected into the mammary fat pad at a 1:1 ratio of matrigel (Corning) and sterile 1 x PBS into 6–8 week old mice of the appropriate genotypes. When the largest tumor reached 1cm3 the mice were euthanized and the tumors were homogenized as above and processed for Ki67 flow cytometry as listed in 'Quantification of Ki67 and DNA content'.

## Agent-based model for impact of mitochondrial transfer on cell division over time

The agent based model performs a Monte-Carlo simulation of individual cell 'agents' over time. At every timepoint, cells increase in mass, $m$, over the simulated time interval $\Delta t$ according to an exponential growth law:

$$m\left(t+1\right) = m\left(t\right) + \mathrm{k} \cdot m\left(t\right) \cdot \Delta t \tag{1}$$

here $k$ is the exponential growth constant. Over every time interval, a fraction of cells, $f_{\Delta t}$, gain transferred mitochondria:

$$f_{\Delta t} = f\frac{\Delta t}{T_d} \tag{2}$$

where $f$ is the overall fraction of the population gaining mitochondria (set to 5% based on our observation that this fraction of the population has transferred mitochondria) over the cell doubling time, $T_d$.

The exponential growth constant, $k$, is then equal to $k_0$, the baseline growth rate, for cell agents without transferred mitochondria, or $k_0 r$, where $r$ is the factor of growth rate increase for cells with transferred mitochondria.

If the mass of a cell is greater than double its baseline mass, then it divides into two new daughter cells, each at half the mass of the parent, that are then tracked in the simulation. We assume that half the population loses transferred mitochondria (and the associated growth increase) with every division.

The mass of the population, $m_P$, is then found by summing the mass of all individual cell agents at a given time.

This result can be compared to the overall final tumor mass in the baseline case, $m_B$, after a given number, $d$, of doublings based on pure exponential growth:

$$m_B = m_0 e^{d \ln 2} \tag{3}$$

This result is plotted in *Figure 5—figure supplement 2* for the case of 5% of the tumor cell population receiving macrophage mitochondria.

## Data and materials availability

The code for QPI analysis is available on GitHub (https://github.com/Zangle-Lab/Macrophage_tumor_mito_transfer).

Single-cell RNA-sequencing data are available in GEO accession number GSE181410. The code for single-cell RNA-sequencing analysis is available on GitHub (https://github.com/rohjohnson-lab/kidwell_casalini_2021; RRID:SCR_002630 (Version 1)).

All other data are available in the main text or in the Supplementary Data.

## Image analysis

All images taken with the Airyscan detector on the Zeiss LSM 880 were subjected to deconvolution using the Zen software (Carl Zeiss) with 'auto' settings (referred to as AiryScan processed). Maximum intensity projections of selected z-planes were generated using Zen or FIJI software (*Schindelin et al.,*

*2012*). Linear adjustments to the brightness and contrast were made using FIJI. Images were cropped and panels were assembled using Adobe Photoshop and Illustrator, respectively (Adobe, Inc).

## Graphical representations and statistical analysis

All graphs were generated using Prism software (v9, GraphPad). All graphs show mean with standard error of the mean. Statistical analyses were performed using both Excel (v16.51, Microsoft) and Prism. Statistical tests used and p-value ranges are indicated in each figure legend. Nested statistical tests were used to take into account the technical replicates within each biological replicate in the analysis of variance tests. Flow cytometry data and representations were analyzed and generated using FlowJo software (v10.7, BD). Welches t-test was used when the goal was to compare mean values of data with normal distribution, and Mann-Whitney analyses was applied when the data was not normally distributed. Two-way ANOVA was utilized when comparing how two independent variables influence a dependent variable. All statistical methodologies were performed under the guidance of biostatistician, Dr. Kenneth M. Boucher.

## Acknowledgements

We thank Wes Sundquist and all members of the Roh-Johnson lab for helpful discussions and edits to this manuscript; ARUP Laboratories for providing leukofilters; James Carrington, Dong Hwi Bae, and Joshua Monts for technical support; Hannah Young for help with data analysis; Alan Aderem and Elizabeth Gold for their mentorship to GSO; and Kenneth M Boucher for help with statistics. We also thank Sadie Johnson for help with animal care, and Kristin Weber Bonk and Ruth Keri for help with animal experiments. We also thank the Huntsman Cancer Institute Cancer Center Shared Resources; the University of Utah Flow Cytometry Core for technical assistance; and the University of Utah Cell Imaging Core for use of the Leica Yokogawa CSU-W1 spinning disc confocal microscope.

## Additional information

### Funding

| Funder | Grant reference number | Author |
| --- | --- | --- |
| National Institutes of Health | R37CA247994 | Minna Roh-Johnson |
| U.S. Department of Defense | W81XWH-20-1-0591 | Minna Roh-Johnson |
| The Mary Kay Foundation | 10-19 | Minna Roh-Johnson |
| National Institutes of Health | R00CA190836 | Chelsea U Kidwell Minna Roh-Johnson |
| National Institutes of Health | F31CA250317 | Joseph R Casalini |
| National Institutes of Health | K99 CA248611 | Defne Bayik |
| National Institutes of Health | TL1 TR002549 | Dionysios C Watson |
| Lerner Research Institute, Cleveland Clinic | | Justin D Lathia |
| Case Comprehensive Cancer Center, Case Western Reserve University | | Justin D Lathia |
| VeloSano Bike Ride | | Defne Bayik Dionysios C Watson Justin D Lathia |
| U.S. Department of Defense | W81XWH1910065 | Thomas A Zangle |

| Funder | Grant reference number | Author |
|---|---|---|
| National Institutes of Health | U54CA224076 | Alana L Welm |
| Breast Cancer Research Foundation | | Alana L Welm |

The funders had no role in study design, data collection and interpretation, or the decision to submit the work for publication.

## Author contributions

Chelsea U Kidwell, Conceptualization, Resources, Data curation, Formal analysis, Supervision, Funding acquisition, Validation, Investigation, Visualization, Methodology, Writing – original draft, Writing – review and editing; Joseph R Casalini, Conceptualization, Resources, Data curation, Formal analysis, Funding acquisition, Validation, Investigation, Visualization, Methodology, Writing – original draft, Project administration, Writing – review and editing; Soorya Pradeep, Conceptualization, Data curation, Formal analysis, Investigation, Methodology; Sandra D Scherer, Jarrod S Johnson, Alana L Welm, Resources, Methodology; Daniel Greiner, Data curation, Software, Formal analysis, Methodology; Defne Bayik, Formal analysis, Validation, Methodology; Dionysios C Watson, Formal analysis, Methodology; Gregory S Olson, Resources; Justin D Lathia, Funding acquisition; Jared Rutter, Writing – review and editing; Thomas A Zangle, Resources, Data curation, Software, Formal analysis, Methodology; Minna Roh-Johnson, Conceptualization, Resources, Data curation, Formal analysis, Supervision, Funding acquisition, Investigation, Visualization, Methodology, Writing – original draft, Project administration, Writing – review and editing

## Author ORCIDs

Chelsea U Kidwell ⓘ http://orcid.org/0000-0003-4269-2503
Joseph R Casalini ⓘ http://orcid.org/0000-0002-3515-3248
Sandra D Scherer ⓘ http://orcid.org/0000-0002-7943-8595
Jarrod S Johnson ⓘ http://orcid.org/0000-0001-7355-5240
Jared Rutter ⓘ http://orcid.org/0000-0002-2710-9765
Minna Roh-Johnson ⓘ http://orcid.org/0000-0003-3961-4547

## Ethics

All animal experiments were approved by the Institutional Animal Care and Use Committee (IACUC) at the University of Utah (PHS Assurance Registration Number: A3031-01; USDA Registration Number: 87-R-0001; protocol #19-12001) and at the Cleveland Clinic (protocol #2179). In accordance to approved protocol, all animals were anesthetized appropriately to assure maximum comfort throughout the duration of procedures. When tumors were grown to approved volumes, mice were humanely euthanized with slow $CO_2$ gas exchange for 5 minutes.

## Decision letter and Author response

Decision letter https://doi.org/10.7554/eLife.85494.sa1
Author response https://doi.org/10.7554/eLife.85494.sa2

# Additional files

## Supplementary files

• MDAR checklist

## Data availability

The code for QPI analysis is available on GitHub (https://github.com/Zangle-Lab/Macrophage_tumor_mito_transfer; copy archived at *ZangleLab, 2023*) for Figure 1.Single-cell RNA-sequencing data are available in GEO accession number GSE181410. The code for single-cell RNA-sequencing analysis is available on GitHub (https://github.com/rohjohnson-lab/kidwell_casalini_2021; copy archived at *rohjohnson-lab, 2023*) for Figure 1.

The following previously published dataset was used:

| Author(s) | Year | Dataset title | Dataset URL | Database and Identifier |
|---|---|---|---|---|
| Roh-Johnson M, Greiner D | 2021 | Macrophage and MDA-MB-231 coculture and mitochondrial transfer | http://www.ncbi.nlm.nih.gov/geo/query/acc.cgi?acc=GSE181410 | NCBI Gene Expression Omnibus, GSE181410 |

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

# Appendix 1

**Appendix 1—key resources table**

| Reagent type (species) or resource | Designation | Source or reference | Identifiers | Additional information |
|---|---|---|---|---|
| strain, strain background (*M. musculus*) | wildtype C57BL/6 J | The Jackson Laboratory | Stock #000664 | |
| strain, strain background (*M. musculus*) | mito:mKate2 mouse Tg(CAG-mKate2)1Poche/J | The Jackson Laboratory | Stock 032188 | |
| strain, strain background (*M. musculus*) | LysM-Cre mouse B6.129P2-Lyz2tm1(cre)Ifo/J | The Jackson Laboratory | Stock 004781 | |
| strain, strain background (*M. musculus*) | Lox-stop-lox-MitoTag mouse B6N.Cg-Gt(ROSA)26Sortm1(CAG-EGFP*)Thm/J | The Jackson Laboratory | Stock 032675 | |
| cell line (*Homo-sapiens*) | MDA-MB-231 | American Type Culture Collection | HTB-26 | |
| cell line (*Homo-sapiens*) | MDA-MB-468 | American Type Culture Collection | HTB-132 | |
| cell line (*Homo-sapiens*) | A375 | American Type Culture Collection | CRL-1619 | |
| cell line (*Homo-sapiens*) | THP-1 | American Type Culture Collection | TIB-202 | |
| cell line (*Homo-sapiens*) | MCF10a | American Type Culture Collection | CRL-10317 | |
| cell line (*M. musculus*) | E0771 | American Type Culture Collection | CRL-3461 | |
| transfected construct (*Homo-sapien*) | pLPCX mito-Grx1-roGFP2 | Addgene | 64977 | |
| transfected construct (*S. cerevisiae*) | pLPCX mito-roGFP2-Orp1 | Addgene | 64992 | |
| antibody | BV711-CD11b (mouse anti-human, monoclonal) | Biolegend | 301344 | 1:20-1:40 Used for flow cytometry |
| antibody | PE anti-human CD326 (EpCAM) Antibody (mouse andti-human, monoclonal) | Biolegend | 369806 | 1:40 Used for flow cytometry |
| antibody | APC-Ki67 (mouse anti-human, Monoclonal) | ThermoFisher | 17-5699-42 | 1:20 – 1:40 Used for flow cytometry |
| antibody | anti-GFP (Chicken, polyclonal) | Abcam | AB13970, | 1:500 |
| antibody | anti-RFP (rabbit, polyclonal) | Abcam | AB62341 | 1:1000 |
| antibody | Alexa Fluor 488 AffiniPure (Goat anti-Chicken, polyclonal) | Jackson ImmunoResearch | 103-545-155 | 1:500 |
| antibody | IgG (H+L) (Cross-Adsorbed Goat anti-Rabbit Alexa Fluor 555, polyclonal) | Invitrogen | A21428 | 1:500 |
| recombinant DNA reagent | Mito-mEm: pLKO.1 mito-mEmerald | This paper | Addgene, 174548 | Lentiviral construct to transfect and express fluorescently-tagged mitochondria |

*Appendix 1 Continued on next page*

*Appendix 1 Continued*

| Reagent type (species) or resource | Designation | Source or reference | Identifiers | Additional information |
|---|---|---|---|---|
| recombinant DNA reagent | Mito-RFP: pLKO.1 mito-TagRFP-T | This paper | Addgene, 174543 | Lentiviral construct to transfect and express fluorescently-tagged mitochondria |
| recombinant DNA reagent | mEmerald-TOMM20: pLKO.1 mEmerald-TOMM20-N-10 | This paper | Addgene, 54282 | Lentiviral construct to transfect and express fluorescently-tagged mitochondria |
| recombinant DNA reagent | mCherry-TOMM20: pLKO.1 mCherry-TOMM20-N-10 | This paper | Addgene, 55146 | Lentiviral construct to transfect and express fluorescently-tagged mitochondria |
| recombinant DNA reagent | Mito-KR: pLKO.1 3xHA-KillerRed-OMP25 | This paper | Addgene, 174544 | Lentiviral construct to transfect and express mitochondrially-localized KillerRed |
| recombinant DNA reagent | ERK-KTR-mRuby: pLentiPGK Blast DEST ERKKTRmRuby2 | Addgene | 90231 | Lentiviral construct to transfect and express ERK Kinase Translocation reporter |
| recombinant DNA reagent | ERK-KTR-Clover: pLentiPGK Puro DEST ERKKTRClover | Addgene | 90227 | Lentiviral construct to transfect and express ERK Kinase Translocation reporter |
| recombinant DNA reagent | Non-target-shRNA | Sigma | SHC002 | Lentiviral construct to transfect and express non-target shRNA |
| recombinant DNA reagent | DRP1-KD: *DRP1*-shRNA | Sigma | TRCN0000001097 | Lentiviral construct to transfect and knock down gene target HGNC ID 2973 |
| sequence-based reagent | Primer: *DRP1*-F | This paper | | AGAAAATGGGGTGGAAGCAGA |
| sequence-based reagent | Primer: *DRP1*-R | This paper | | AAGTGCCTCTGATGTTGCCA |
| sequence-based reagent | Primer: *GAPDH-F* | This paper | | AGCCACATCGCTCAGACA |
| sequence-based reagent | Primer: *GAPDH-R* | This paper | | ACATGTAAACCATGTAGTTGAGGT |
| peptide, recombinant protein | GM-CSF | Peprotech | 300–03 | 20 ng/mL |
| peptide, recombinant protein | IFN-γ | Peprotech | 3000–02 | 20 ng/mL |
| peptide, recombinant protein | IL-4 | Peprotech | 200–04 | 20 ng/mL |
| peptide, recombinant protein | IL-13 | Peprotech | 200–13 | 20 ng/mL |
| commercial assay or kit | eBioscience Foxp3/ Transcription Factor Staining Buffer Set | ThermoFisher | 00-5523-00 | |
| commercial assay or kit | Polyplus-transfection jetPRIME DNA/siRNA transfection kit | Genesee Scientific | 55–131 | Used to transfect pLPCX mito-Grx1-roGFP2 and pLPCX mito-roGFP2-Orp1 probes |
| commercial assay or kit | MitoTracker Deep Red | ThermoFisher | M22426 | Used at 25 nM |
| commercial assay or kit | TMRM: Tetramethylrhodamine, Methyl Ester, Perchlorate | ThermoFisher | T668 | Used at 100 nM |
| commercial assay or kit | LysoTracker Blue | ThermoFisher | L7525 | Used at 75 nM |
| commercial assay or kit | MemBrite 640/660 | Biotium | 30097 | Used at 1:1000 |

*Appendix 1 Continued*

| Reagent type (species) or resource | Designation | Source or reference | Identifiers | Additional information |
|---|---|---|---|---|
| commercial assay or kit | DCFDA: Carboxy-H2DCFDA | ThermoFisher | C400 | Used at 5µM |
| chemical compound, drug | ERKi: SCH772984 | SelleckChem | 7101 | Used at 1 µM |
| chemical compound, drug | PMA: Phorbol 12-myristate 13-acetate | SelleckChem | S7791 | Used at 100 nM (cancer cell treatment) and 162 nM (THP1 differentiation) |
| chemical compound, drug | MitoTEMPO | Cayman Chemical | 16621 | Used at 100 µM |
| software, algorithm | QPI analyses | This Paper | | https://github.com/Zangle-Lab/Macrophage_tumor_mito_transfer |
| software, algorithm | Single-cell RNA-sequencing | This paper | | GEO accession number: GSE181410 (RRID:SCR_002630 (version number 1)) https://github.com/rohjohnson-lab/kidwell_casalini_2021 |

