## [Editor Report]

This important work demonstrates that the transfer of dysfunctional mitochondria stimulates proliferation in recipient cancer cells by serving as a signal to induce reactive oxygen species production that in turn activates signaling pathways that control cell cycle. Compelling cell biology assays including rigorous microscopy with elegant reporters track the function and fate of transferred mitochondria in recipient cells. The work is relevant to the study of mitochondria, cancer, and immune cells and will be of broad interest to cell biologists and biochemists.

---

## [Decision Letter]

**Decision letter after peer review:**

Thank you for submitting your article "Transferred mitochondria act as a signaling source promoting proliferation" for consideration by *eLife*. Your article has been reviewed by 3 peer reviewers, one of whom is a member of our Board of Reviewing Editors, and the evaluation has been overseen by Benoît Kornmann as the Senior Editor.

As you will see below, the reviewers found the work to be of potential interest but raised significant concerns that would require additional experimentation prior to consideration of a revised version. The reviewers have opted to remain anonymous.

All three reviewers found the paper to have interesting implications for many fields-- from mitochondrial biology to cancer biology and beyond-- and to employ multiple convincing methods to demonstrate transfer of mitochondria with potential to alter recipient cell function. At the same time, all reviewers raised similar concerns about the strength and validity of the proposed model. A key question raised by all 3 reviewers is how the transferred mitochondria produce ROS in the absence of a membrane potential. A related concern is that mitoTEMPO, the antioxidant used to show that quenching ROS dissipates the pro-mitogenic signal of transferred mitochondria, relies on membrane potential to accumulate within mitochondria. Therefore, whether the ROS are made specifically by transferred mitochondria to control cell signaling is not fully demonstrated at this time. Revisions to address these central questions are discussed below. The reviewers have discussed their reviews with one another, and the Reviewing Editor has drafted this to help you prepare a revised submission.

Essential revisions:

Overall, the reviewers each shared several major concerns. For completeness, I am including the entire reviewer remarks below to provide full context for all requested revision. Here, I summarize the three key essential revision requirements alongside other major/minor points that should be addressed. Reviewer points where data would be welcome, but not required, are specifically noted at the end.

Essential questions:

1. How can defective mitochondria produce ROS? If they do not have membrane potential, are they respiring? If not, do they have functional electron transfer reactions that can generate ROS? As mitoTEMPO accumulates in mitochondria as a result of membrane potential, the relevance of these experiments are not clear. (Please see full comments below for more reviewer context.)

2. A related questions is whether ROS is generated specifically by the transferred mitochondria? Use of mitoTEMPO in quenching ROS may illustrate that the ROS is not coming from transferred mitochondria, or the transferred mitochondria regain membrane potential, as this antioxidant like other triphenylphosphonium-based antioxidants accumulate based on membrane potential. The authors partially address this with barcoded biosensors, but this hampers the ROS-arguments in the paper, and the authors should readdress interpretations. Related, ROS quenching and ERK inhibition only affect proliferation when it is done in transferred mitochondria. Why don't ROS and ERK affect proliferation in control cells? (See comments by Reviewer #1 and #2, below.)

3. How much of the transferred mitochondria and/or mitochondrial ROS is beneficial? (See full comment from Reviewer #3 below.)

Other essential reviewer comments to address:

– Figure 1c. Why does the recipient cell need to be cancerous (referring to the MCF10A experiment). Is there sufficient experimental evidence for this claim?

– Figure 1 – Supplement 2a. In line with ROS signaling in the following experiments – were the proliferative program observed in Figure 1 – Sup 2a ROS dependent?

– The authors show in Figure 2 that it takes 48 h for mitochondria to become oxidized. In Figure 1 it appears that the proliferation benefit occurs as early as 24 h of coculture. How do the authors reconcile these timescales?

*Reviewer #1 (Recommendations for the authors):*

Questions:

– One major question is how do mitochondria without membrane potential induce ROS? Do they have functional electron transfer reactions that are producing ROS?

– The authors show in Figure 2 that it takes 48 h for mitochondria to become oxidized. In Figure 1 it appears that the proliferation benefit occurs as early as 24 h of coculture. How do the authors reconcile these timescales?

­

– I'm puzzled by the observation that ROS quenching and ERK inhibition only affect proliferation when it is done in transferred mitochondria. Why don't ROS and ERK affect proliferation in control cells?

*Reviewer #2 (Recommendations for the authors):*

The manuscript will be strengthened if the following substantial revisions based on the following points.

Methods

The authors include sufficient experimental detail and utilize appropriate model systems.

Results

Figure 1c. Why does the recipient cell need to be cancerous (referring to the MCF10A experiment). Is there sufficient experimental evidence for this claim. Can the authors speculate on the regulation of mitochondrial uptake by cancer cells, specifically, in vitro?

Figure 1g. The authors aim to isolate mitochondria from macrophages and test the ability of isolated mitochondria in promoting proliferation in cancer cell lines. The authors performed a crude mitochondrial isolation protocol here, without sucrose or percoll gradient ultracentrifugation to purify mitochondria from associated endoplasmic reticulum. The authors do not illustrate the extent of purification of the isolated mitochondria. This makes the data difficult to interpret, and the effects of ER contamination on proliferation are not controlled for. We suggest the authors perform this experiment with an ultracentrifuge-purified sample. Alternatively, mito-tag approaches can be used to IP intact mitochondria from cells.

Figure 1 – Supplement 2a. The authors perform single-cell RNAseq in MDA-MB-231 +/- macrophage mitochondria. Can the authors comment on the duration of mitochondrial uptake in these cells prior to RNAseq? Is it an acute proliferative response (gene expression program) to the uptake of mitochondria (short-term), or a robust adaptation (long-term). The authors indicate that the daughter cells exhibit sustained increases in growth rates – can the authors then indicate the time of RNAseq? Also, in line with ROS signaling in the following experiments – were the proliferative program observed in Figure 1 – Sup 2a ROS dependent?

Figure 2. General comment regarding transferred mitochondria without membrane potential. Are these mitochondria respiring? Where is the source of ROS coming from, and how is the membrane potential dissipated if these mitochondria are in fact respiring? Is this through reverse ATP synthase activity as it occurs in mitochondrial deleted cells? Can the authors illustrate the capacity at which these macrophage-mitochondria are respiring? Can the authors show that the ROS is specific to the transferred mitochondria and not an off-target ROS generation? Use of mitoTEMPO in quenching ROS may illustrate that the ROS is not coming from your transferred mitochondria, or the transferred mitochondria regain membrane potential. This antioxidant, and other triphenylphosphonium-based antioxidants accumulate based on membrane potential. The authors partially address this with barcoded biosensors, but this hampers the ROS-arguments in the paper, and the authors should readdress interpretations.

Figure 3. Similar as figure 2. The authors illustrate activation of ERK by photobleaching induced ROS. Although this illustrates that ROS activates ERK in this system, the authors do not illustrate that this ROS originates from macrophage mitochondria. As the authors mention, it is known that ROS induces ERK signaling. This section appears to further validate known mechanisms of ERK activation without connecting to the transferred mitochondria.

Figure 4. The authors illustrate that M2-macrophages exhibit enhanced mitochondrial transfer efficiency than M1. This supports the notion that M2 macrophages are pro-tumorigenic, here by a novel mechanism. Further data illustrating mitochondrial fission as an important regulator of the transfer process, as DRP1 knockdown with shRNA attenuates transfer.

*Reviewer #3 (Recommendations for the authors):*

Overall, the paper is convincing of its primary points, but expounding on some aspects is warranted to contextualize these findings with respect to the dose response of the effects and to address other studies on mitochondrial transfer in the literature:

1) How much transferred mitochondria and/or mitochondrial ROS is beneficial?

While the benefits of ROS in some contexts have been increasingly appreciated, ROS are also well known to cause deleterious effects at higher doses. The authors conclude that transferred mitochondria promote proliferation through mitochondrial ROS activation of ERK, but some understanding of the boundaries of this effect would help with interpretation. Some potential approaches to address this issue could include: Are there dose responsive effects with respect to proliferation or ERK signaling amongst cells with different numbers or volumes of transferred mitochondria? Is there a dose response to the cell cycle changes from macrophage mitochondria bath dosages corresponding to the number of mitochondria transferred? What range of mito-KillerRed activation is beneficial, and at what point is it deleterious?

2) How do we reconcile the finding that transferred macrophage mitochondria are persistently dysfunctional?

As the authors noted, the literature has focused on situations where mitochondrial transfer can repopulate functional mitochondria in cancer cells with endogenously dysfunctional mitochondria, but here the authors conclude that transferred mitochondria are persistently dysfunctional in recipient cells. While this discrepancy is not likely to be entirely solved here, some simple experiments could help put context on the salient differences between studies by addressing how their biological system might influence the results: Can transfer of macrophage mitochondria reconstitute mitochondrial function of rho zero cells? Does transfer of mitochondria amongst 231 cell populations (e.g.. transfer of 231 cell mitochondria expressing mito-RFP to 231 cells expressing mito-mEmerald) also result in persistently dysfunctional mitochondria in recipient cells? Do mEmerald expressing macrophages have functional mitochondria prior to mitochondrial transfer?

---

## [Author Response]

Essential revisions:Essential questions:1. How can defective mitochondria produce ROS? If they do not have membrane potential, are they respiring? If not, do they have functional electron transfer reactions that can generate ROS? As mitoTEMPO accumulates in mitochondria as a result of membrane potential, the relevance of these experiments are not clear. (Please see full comments below for more reviewer context.)

We cannot say for certain that transferred mitochondria generate the ROS or if it is generated elsewhere in the cell. However, our rationale for the mito-KillerRed experiments is to experimentally test whether inducing small amounts of ROS is sufficient to promote cell division, regardless of the source. We used mitochondrially localized KillerRed as we wanted to control for the photobleaching procedure itself – ie. if KillerRed were spatially localized, then we can photobleach a KillerRed-negative region as a control. It is not possible to quantify the exact amount of ROS in cancer cells with macrophage mitochondria, but our experiments show that cancer cells with macrophage mitochondria have increased intracellular ROS, the biosensor experiments suggest that ROS is in a localized region at transferred mitochondria (Figure 2c,d), and that quenching ROS specifically inhibits proliferation of cancer cells with macrophage mitochondria, without affecting cancer cell proliferation more broadly (Figure 2f). From these results, we aimed to produce small amounts of ROS by photobleaching small 2µm x 2µm regions of interest, and determining whether this small amount of ROS production is sufficient to cause cell division, regardless of where the ROS is generated. If we performed similar experiments with cytoplasmic KillerRed (not mitochondrially-localized), we would expect the same response, although the negative control photobleaching would be much harder to perform, which is why we opted to use the mitochondrially-localized KillerRed. The outcome of this experiment is that inducing small amounts of ROS is sufficient to induce proliferation.

Second, it appears that the mito-Grx1-roGFP2 sensors are encoded in the recipient cells; how do these sensors then become localized to the transferred mitochondria?

This is a great question, and one that we have thought about as well. Yes, the mito-Grx1-roGFP2 sensor is encoded in the recipient cells. The construct contains an MTS from an ATP synthase that targets it to the mitochondria. Due to the reduced membrane potential, we do not know whether the sensor is imported into the mitochondria, although it is possible that some of the mitoGrx1-roGFP2 sensor is inserted into the transferred mitochondria, if the transferred mitochondria retain very low levels of membrane potential (low enough to not be detected by the membrane potential-sensitive dyes). Regardless of whether the sensor is imported into the mitochondria, we interpret our findings to suggest that the Grx1-roGFP2 sensor is targeted to the transferred mitochondria, and the sensor is reading out the oxidized versus reduced forms of Grx1 at that site, even if it is not imported into the mitochondria. Our understanding from the literature and from discussions with our metabolism colleagues is that the sensor does not need to be inserted into the transferred mitochondria to show the redox state of Grx1 in the vicinity of the transferred mitochondria, and that the strength of protein insertion is dependent on the specific MTS and the protein it is fused to (Schafer, Bozkurt et al. 2022). For example, PINK contains an MTS, and can still localize and be processed by mitochondria in the absence of membrane potential (Becker, Richter et al. 2012). PINK1 can also be retained at the outer mitochondrial membrane in the absence of an MTS altogether, thereby suggesting that mitochondrial insertion is not required for maintained mitochondrial localization (Zhou, Huang et al. 2008, Liu, Vives-Bauza et al. 2009, Weihofen, Thomas et al. 2009). Our work shows that reactive oxygen species accumulate at the site of transferred mitochondria using two different biosensors. We think that the biosensor is constantly in flux, with a constant flow of Grx1-roGFP2 targeted to mitochondria, and becoming oxidized at the transferred mitochondria. We tried to test this hypothesis by performing FRAP (fluorescence recovery after photobleaching) experiments of the biosensor at transferred mitochondria and quantifying fluorescence recovery over time. We observed recovery of the biosensor at transferred mitochondria, with the biosensor primarily showing the oxidized version of Grx1 at this site, but due to the dynamic nature of the mitochondria, the transferred mitochondria were hard to track without overlapping with endogenous mitochondria, which distorted the quantification. As a result, we could not reliably quantify the FRAP. However, we have included representative images at time points from these experiments before and after photobleaching. The cancer cell is expressing mito-Grx1-roGFP2. Macrophage mitochondria are red, and an arrowhead marks the macrophage mitochondria that we followed after photobleaching. The zoom panels in Author response image 1 highlights the macrophage mitochondria targeted for photobleaching as a merged image (showing red transferred mitochondria outlined in gray; yellow oxidized-Grx1; and green reduced-Grx1), and then an oxidized-Grx1 only panel to better visualize this state. Before photobleaching the sensor, we can visualize the oxidized form of mito-Grx-roGFP2 at the site of transferred mitochondria in “A”. We then photobleached the sensor at transferred mitochondria in “B”. We know that we photobleached the sensor at this site, as we visualized decreased levels of oxidized Grx1-roGFP2 in the right-most panel of “B”. 40 seconds after photobleaching, the sensor returns to the site of transferred mitochondria in “C”. To ensure that the oxidized form of mito-Grx1-roGFP2 is tracking with the transferred macrophage mitochondria, we also looked another 5 seconds later, and can still visualize enriched oxidized Grx1 at that site in “D”. We interpret these preliminary results to suggest that the sensor is in flux and can continually target to transferred mitochondria.

**Author response image 1. sa2fig1:** 

Thus, while we cannot comment on whether or how much of the sensor is inserted into the transferred mitochondria, nor can we perform experiments to quantify the extent of Grx1-roGFP2 processing by these mitochondria due to the lack of protein input for biochemical assays, we interpret these data as ROS accumulates at the site of transferred mitochondria, which in itself was an unexpected and exciting discovery, and a central finding of the work.

2. A related questions is whether ROS is generated specifically by the transferred mitochondria? Use of mitoTEMPO in quenching ROS may illustrate that the ROS is not coming from transferred mitochondria, or the transferred mitochondria regain membrane potential, as this antioxidant like other triphenylphosphonium-based antioxidants accumulate based on membrane potential. The authors partially address this with barcoded biosensors, but this hampers the ROS-arguments in the paper, and the authors should readdress interpretations. Related, ROS quenching and ERK inhibition only affect proliferation when it is done in transferred mitochondria. Why don't ROS and ERK affect proliferation in control cells? (See comments by Reviewer #1 and #2, below.)

Regarding the ROS quenching and ERK results, we specifically sought quenching and inhibitor concentrations that inhibit/quench the target, but have little effect on cancer cell monoculture proliferation. We performed these critical experiments to specifically test whether ROS/ERK inhibition affects the proliferative capacity of cancer cells *with macrophage mitochondria*, as opposed to proliferation generally. We show representative ERKi data in Author response image 2 – We tested 5 different ERKi concentrations, and evaluated both inhibition of ERK activity (as measured by flow cytometry quantifying the activated phosphorylated ERK, p-ERK, in “Author response image 2”), as well as changes to the G2/M phase of the cell cycle (black bar in graph in “Author response image 2”). These preliminary results showed that both 0.3µM and 1µM ERKi inhibited ERK activity, but did not affect the G2/M phase of the cell cycle in cultured 231 cancer cells alone (n=1 biological replicate; n=3 technical replicates). Thus, we used the 1 µM ERKi for subsequent coculture assays in the manuscript (Figure 3e,f; each with n=3 biological replicates). We have made modifications to the text to better articulate this important point.

3. How much of the transferred mitochondria and/or mitochondrial ROS is beneficial? (See full comment from Reviewer #3 below.)

This is an interesting question. We have addressed this question by performing some of the experiments suggested by this reviewer. To determine whether there are dose responsive effects with respect to proliferation or ERK signaling amongst cells with different amounts of transferred mitochondria, we analyzed populations of cancer cells with either “high” or “low” amounts of transferred macrophage mitochondria. These “high” and “low” designations were determined by taking into the account the median FITC fluorescence intensity for cancer cells with transferred mitochondria, while maintaining equal numbers of cancer cells in both the “high” and “low” population to avoid calculating percentages on small numbers of cells (Author response image 3 for representative gating). We then analyzed the percentages of cells in the different phases of the cell cycle (Author response image 3, statistics shown specifically for the proliferative G2/M phase of the cell cycle). Both the “high” and “low” populations showed an increase in the percentage of cells in the G2/M phase of the cell cycle compared to the “no transfer” population, consistent with what we report in the manuscript, but we found no difference *between* the “high” and “low” populations suggesting that the amount of macrophage mitochondrial transfer to cancer cells observed in our system did not affect cell cycle stages in a dose-dependent manner.

**Author response image 3. sa2fig3:** A. Representative gating strategy for “low” vs “high” macrophage mitochondrial transfer in macrophage/cancer cell cocultures. B. Cell cycle analysis for populations in (A). 2-way ANOVA, n=2 biological replicates, p<0.001. C. ERK-KTR analysis with imagestream, n=3 biological replicates, 1-way ANOVA, p<0.0001. D. Scatterplot of ERK-KTR and mitochondrial transfer mean fluorescence intensity (MFI). Each dot is a cell. n=7,966 cells. Correlation R^2^ value: 3.261e-005. E. Representative gating strategy for “low” vs “high” macrophage mitochondrial uptake by cancer cells. F. Cell cycle analysis for populations in (E). 2-way ANOVA, n=3 biological replicates, p<0.001.

We then performed similar analysis for ERK activation, in which we used the ERK-KTR biosensor to measure the cytoplasmic/nuclear ratios of ERK, with higher cytoplasmic/nuclear ratios indicating higher ERK activity. Similar to our proliferation results, we did not observe differences in ERK activity between the “high” and “low” mitochondrial transfer populations (Author response image 3). With the ERK-KTR biosensor, we are also able to quantify ERK activity on a cell-by-cell basis, thus we analyzed each cell and graphed ERK activity (y-axis) with respect the fluorescence intensity of mitochondrial transfer in the same cell (x-axis) (Author response image 3). We did not observe a positive correlation between the amount of mitochondrial transfer and ERK activity (R^2^ value: 3.261e-005).

Finally, we determined whether there was a dose response to cell cycle changes with macrophage mitochondrial bath applications. Similar to our analysis with cocultures in Figure A, we divided the cancer cell population into “high” vs “low” mitochondrial uptake (Figure E), and analyzed cell cycle stages (Figure F). We did not observe dose-dependent cell cycle changes in response to macrophage mitochondrial uptake.

Together, these results suggest that cancer cells with macrophage mitochondria exhibit increased ERK activity and increased proliferation, and that the increased ERK activity and proliferation does not show a dose-dependent response to mitochondrial transfer. We hypothesize that the mitochondrial transfer, and the subsequent ROS accumulation, that we observe within this system show beneficial (proliferation-promoting) effects, and is not within the boundaries of inducing deleterious effects.

Consistent with this hypothesis, previous reports using KillerRed either induce cell death or generate phototoxic damage (Bulina, Chudakov et al. 2006, Li, Fang et al. 2019). These approaches typically use high laser settings, irradiate the whole cell, and use long and continuous illumination times in the order of several minutes (Bulina, Chudakov et al. 2006, Williams, Bejjani et al. 2013, Kuznetsova, Shirmanova et al. 2015, Li, Fang et al. 2019). Our approaches with mito-KillerRed use small, 2 m x 2 m, regions of interest and illumination on the 5-10 seconds time scale, and thus suggest that we are likely generating low levels of ROS (as indicated by Figure 3 – Figure S5), and not near the ROS levels that induce cell death.

We did not include these data into the manuscript as we were not clear what additional insights the findings added to the work given that we did not observe a dose-dependent effect, however we would be happy to do so, if the reviewers prefer.

Other essential reviewer comments to address:– Figure 1c. Why does the recipient cell need to be cancerous (referring to the MCF10A experiment). Is there sufficient experimental evidence for this claim?

We do not have evidence that the recipient cell needs to be cancerous to receive macrophage mitochondria. The MCF10A result shows that macrophages exhibit decreased transfer efficiencies to MCF10A than the other three malignant cell lines we tested. We do think that macrophage mitochondrial transfer can occur to non-malignant cell lines as illustrated in other published work (Nicolas-Avila, Lechuga-Vieco et al. 2020, Brestoff, Wilen et al. 2021, Liu, Wu et al. 2022, van der Vlist, Raoof et al. 2022, Yang, Yokomori et al. 2022), but it is not clear whether the mechanism of mitochondrial dysfunction and ROS accumulation occurs in these systems. We have adjusted the language in the text to reflect this point.

– Figure 1 – Supplement 2a. In line with ROS signaling in the following experiments – were the proliferative program observed in Figure 1 – Sup 2a ROS dependent?

The single cell RNA sequencing results were unfortunately not performed in the presence of mitoTEMPO. While we are unable to answer this specific question with the scRNA sequencing results, we were able to show complementary experiments that the proliferative response observed in cancer cells with macrophage mitochondria can be ameliorated with the addition of mitoTEMPO with a cell-based assay (Figure 2f).

– The authors show in Figure 2 that it takes 48 h for mitochondria to become oxidized. In Figure 1 it appears that the proliferation benefit occurs as early as 24 h of coculture. How do the authors reconcile these timescales?

We apologize if it wasn’t clear. The result in Figure 2D with the mito-Grx1-roGRP2 was within 24 hours, which is the same time frame as the proliferation phenotype (also at 24 hours). We have made the time point clearer in the figure legend.

References:

Akhter, W., J. Nakhle, L. Vaillant, G. Garcin, C. Le Saout, M. Simon, C. Crozet, F. Djouad, C. Jorgensen, M. L. Vignais and J. Hernandez (2023). "Transfer of mesenchymal stem cell mitochondria to CD4(+) T cells contributes to repress Th1 differentiation by downregulating T-bet expression." Stem Cell Res Ther 14(1): 12.

Becker, D., J. Richter, M. A. Tocilescu, S. Przedborski and W. Voos (2012). "Pink1 kinase and its membrane potential (Deltapsi)-dependent cleavage product both localize to outer mitochondrial membrane by unique targeting mode." J Biol Chem 287(27): 22969-22987.

Brestoff, J. R., C. B. Wilen, J. R. Moley, Y. Li, W. Zou, N. P. Malvin, M. N. Rowen, B. T. Saunders, H. Ma, M. R. Mack, B. L. Hykes, Jr., D. R. Balce, A. Orvedahl, J. W. Williams, N. Rohatgi, X. Wang, M. R. McAllaster, S. A. Handley, B. S. Kim, J. G.

Doench, B. H. Zinselmeyer, M. S. Diamond, H. W. Virgin, A. E. Gelman and S. L. Teitelbaum (2021). "Intercellular Mitochondria Transfer to Macrophages Regulates White Adipose Tissue Homeostasis and Is Impaired in Obesity." Cell Metab 33(2): 270-282 e278.

Bulina, M. E., D. M. Chudakov, O. V. Britanova, Y. G. Yanushevich, D. B. Staroverov, T. V. Chepurnykh, E. M. Merzlyak, M. A. Shkrob, S. Lukyanov and K. A. Lukyanov (2006). "A genetically encoded photosensitizer." Nat Biotechnol 24(1): 95-99. Cowan, D. B., R. Yao, V. Akurathi, E. R. Snay, J. K. Thedsanamoorthy, D. Zurakowski, M. Ericsson, I. Friehs, Y. Wu, S. Levitsky, P. J. Del Nido, A. B. Packard and J. D. McCully (2016). "Intracoronary Delivery of Mitochondria to the Ischemic Heart for Cardioprotection." PLoS One 11(8): e0160889.

Crewe, C., J. B. Funcke, S. Li, N. Joffin, C. M. Gliniak, A. L. Ghaben, Y. A. An, H. A. Sadek, R. Gordillo, Y. Akgul, S. Chen, D. Samovski, P. Fischer-Posovszky, C. M. Kusminski, S. Klein and P. E. Scherer (2021). "Extracellular vesicle-based interorgan transport of mitochondria from energetically stressed adipocytes." Cell Metab 33(9): 1853-1868.e1811.

Dong, L. F., J. Kovarova, M. Bajzikova, A. Bezawork-Geleta, D. Svec, B. Endaya, K. Sachaphibulkij, A. R. Coelho, N. Sebkova, A. Ruzickova, A. S. Tan, K. Kluckova, K. Judasova, K. Zamecnikova, Z. Rychtarcikova, V. Gopalan, L. Andera, M. Sobol, B. Yan, B. Pattnaik, N. Bhatraju, J. Truksa, P. Stopka, P. Hozak, A. K. Lam, R. Sedlacek, P. J. Oliveira, M. Kubista, A. Agrawal, K. Dvorakova-Hortova, J. Rohlena, M. V. Berridge and J. Neuzil (2017). "Horizontal transfer of whole mitochondria restores tumorigenic potential in mitochondrial DNA-deficient cancer cells." *ELife* 6.

Feng, Y., Y. Liu, X. Ma, L. Xu, D. Ding, L. Chen, Z. Wang, R. Qin, W. Sun and H. Chen (2022). "Intracellular marriage of bicarbonate and Mn ions as "immune ion reactors" to regulate redox homeostasis and enhanced antitumor immune responses." J Nanobiotechnology 20(1): 193.

Franco-Iborra, S., T. Cuadros, A. Parent, J. Romero-Gimenez, M. Vila and C. Perier (2018). "Defective mitochondrial protein import contributes to complex I-induced mitochondrial dysfunction and neurodegeneration in Parkinson's disease." Cell Death Dis 9(11): 1122.

Frezza, C., S. Cipolat and L. Scorrano (2007). "Organelle isolation: functional mitochondria from mouse liver, muscle and cultured fibroblasts." Nat Protoc 2(2): 287-295.

Korpershoek, J. V., M. Rikkers, F. S. A. Wallis, K. Dijkstra, M. Te Raa, P. de Knijff, D. B. F. Saris and L. A. Vonk (2022). "Mitochondrial Transport from Mesenchymal Stromal Cells to Chondrocytes Increases DNA Content and Proteoglycan Deposition in vitro in 3D Cultures." Cartilage: 19476035221126346.

Kuznetsova, D. S., M. V. Shirmanova, V. V. Dudenkova, P. V. Subochev, I. V. Turchin, E. V. Zagaynova, S. A. Lukyanov, B. E. Shakhov and V. A. Kamensky (2015). "Photobleaching and phototoxicity of KillerRed in tumor spheroids induced by continuous wave and pulsed laser illumination." J Biophotonics 8(11-12): 952-960.

Leadsham, J. E., G. Sanders, S. Giannaki, E. L. Bastow, R. Hutton, W. R. Naeimi, M. Breitenbach and C. W. Gourlay (2013). "Loss of cytochrome c oxidase promotes RAS-dependent ROS production from the ER resident NADPH oxidase, Yno1p, in yeast." Cell Metab 18(2): 279-286.

Li, X., F. Fang, Y. Gao, G. Tang, W. Xu, Y. Wang, R. Kong, A. Tuyihong and Z. Wang (2019). "ROS Induced by KillerRed Targeting Mitochondria (mtKR) Enhances Apoptosis Caused by Radiation via Cyt c/Caspase-3 Pathway." Oxid Med Cell Longev 2019: 4528616.

Liu, W., C. Vives-Bauza, R. Acin-Perez, A. Yamamoto, Y. Tan, Y. Li, J. Magrane, M. A. Stavarache, S. Shaffer, S. Chang, M. G. Kaplitt, X. Y. Huang, M. F. Beal, G. Manfredi and C. Li (2009). "PINK1 defect causes mitochondrial dysfunction, proteasomal deficit and α-synuclein aggregation in cell culture models of Parkinson's disease." PLoS One 4(2): e4597.

Liu, Y., M. Wu, C. Zhong, B. Xu and L. Kang (2022). "M2-like macrophages transplantation protects against the doxorubicin-induced heart failure via mitochondrial transfer." Biomater Res 26(1): 14.

Nakai, M., A. Mori, A. Watanabe and Y. Mitsumoto (2003). "1-methyl-4-phenylpyridinium (MPP+) decreases mitochondrial oxidation-reduction (REDOX) activity and membrane potential (Deltapsi(m)) in rat striatum." Exp Neurol 179(1): 103-110.

Nicolas-Avila, J. A., A. V. Lechuga-Vieco, L. Esteban-Martinez, M. Sanchez-Diaz, E. Diaz-Garcia, D. J. Santiago, A. RubioPonce, J. L. Li, A. Balachander, J. A. Quintana, R. Martinez-de-Mena, B. Castejon-Vega, A. Pun-Garcia, P. G. Traves, E. Bonzon-Kulichenko, F. Garcia-Marques, L. Cusso, A. G. N, A. Gonzalez-Guerra, M. Roche-Molina, S. Martin-Salamanca, G. Crainiciuc, G. Guzman, J. Larrazabal, E. Herrero-Galan, J. Alegre-Cebollada, G. Lemke, C. V. Rothlin, L. J. Jimenez-Borreguero, G. Reyes, A. Castrillo, M. Desco, P. Munoz-Canoves, B. Ibanez, M. Torres, L. G. Ng, S. G. Priori, H. Bueno, J. Vazquez, M. D. Cordero, J. A. Bernal, J. A. Enriquez and A. Hidalgo (2020). "A Network of Macrophages Supports Mitochondrial Homeostasis in the Heart." Cell 183(1): 94-109 e123.

Patel, S. P., F. M. Michael, J. L. Gollihue, W. Brad Hubbard, P. G. Sullivan and A. G. Rabchevsky (2023). "Delivery of mitoceuticals or respiratory competent mitochondria to sites of neurotrauma." Mitochondrion 68: 10-14.

Phinney, D. G., M. Di Giuseppe, J. Njah, E. Sala, S. Shiva, C. M. St Croix, D. B. Stolz, S. C. Watkins, Y. P. Di, G. D. Leikauf, J. Kolls, D. W. Riches, G. Deiuliis, N. Kaminski, S. V. Boregowda, D. H. McKenna and L. A. Ortiz (2015). "Mesenchymal stem cells use extracellular vesicles to outsource mitophagy and shuttle microRNAs." Nat Commun 6: 8472.

Saha, T., C. Dash, R. Jayabalan, S. Khiste, A. Kulkarni, K. Kurmi, J. Mondal, P. K. Majumder, A. Bardia, H. L. Jang and S. Sengupta (2021). "Intercellular nanotubes mediate mitochondrial trafficking between cancer and immune cells." Nat Nanotechnol.

Schafer, J. A., S. Bozkurt, J. B. Michaelis, K. Klann and C. Munch (2022). "Global mitochondrial protein import proteomics reveal distinct regulation by translation and translocation machinery." Mol Cell 82(2): 435-446 e437.

Tan, A. S., J. W. Baty, L. F. Dong, A. Bezawork-Geleta, B. Endaya, J. Goodwin, M. Bajzikova, J. Kovarova, M. Peterka, B. Yan, E. A. Pesdar, M. Sobol, A. Filimonenko, S. Stuart, M. Vondrusova, K. Kluckova, K. Sachaphibulkij, J. Rohlena, P. Hozak, J. Truksa, D. Eccles, L. M. Haupt, L. R. Griffiths, J. Neuzil and M. V. Berridge (2015). "Mitochondrial genome acquisition restores respiratory function and tumorigenic potential of cancer cells without mitochondrial DNA." Cell Metab 21(1): 81-94.

van der Vlist, M., R. Raoof, H. Willemen, J. Prado, S. Versteeg, C. Martin Gil, M. Vos, R. E. Lokhorst, R. J. Pasterkamp, T. Kojima, H. Karasuyama, W. Khoury-Hanold, L. Meyaard and N. Eijkelkamp (2022). "Macrophages transfer mitochondria to sensory neurons to resolve inflammatory pain." Neuron 110(4): 613-626 e619.

Weihofen, A., K. J. Thomas, B. L. Ostaszewski, M. R. Cookson and D. J. Selkoe (2009). "Pink1 forms a multiprotein complex with Miro and Milton, linking Pink1 function to mitochondrial trafficking." Biochemistry 48(9): 2045-2052. Williams, D. C., R. E. Bejjani, P. M. Ramirez, S. Coakley, S. A. Kim, H. Lee, Q. Wen, A. Samuel, H. Lu, M. A. Hilliard and M. Hammarlund (2013). "Rapid and permanent neuronal inactivation in vivo via subcellular generation of reactive oxygen with the use of KillerRed." Cell Rep 5(2): 553-563.

Wu, T. H., E. Sagullo, D. Case, X. Zheng, Y. Li, J. S. Hong, T. TeSlaa, A. N. Patananan, J. M. McCaffery, K. Niazi, D. Braas, C. M. Koehler, T. G. Graeber, P. Y. Chiou and M. A. Teitell (2016). "Mitochondrial Transfer by Photothermal Nanoblade Restores Metabolite Profile in Mammalian Cells." Cell Metab 23(5): 921-929.

Yang, C., R. Yokomori, L. H. Chua, S. H. Tan, D. Q. Tan, K. Miharada, T. Sanda and T. Suda (2022). "Mitochondria transfer mediates stress erythropoiesis by altering the bioenergetic profiles of early erythroblasts through CD47." J Exp Med 219(12).

Zhao, R. Z., S. Jiang, L. Zhang and Z. B. Yu (2019). "Mitochondrial electron transport chain, ROS generation and uncoupling (Review)." Int J Mol Med 44(1): 3-15.

Zhou, C., Y. Huang, Y. Shao, J. May, D. Prou, C. Perier, W. Dauer, E. A. Schon and S. Przedborski (2008). "The kinase domain of mitochondrial PINK1 faces the cytoplasm." Proc Natl Acad Sci U S A 105(33): 12022-12027.